# Vertically nested LES for high-resolution simulation of the surface layer in PALM (version 5.0)

Sadiq Huq[1], Frederik De Roo[1,4], Siegfried Raasch[3], and Matthias Mauder[1,2]

[1]Institute of Meteorology and Climate Research, Atmospheric Environmental Research (IMK-IFU), Karlsruhe Institute of Technology (KIT), Kreuzeckbahnstrasse 19, 82467 Garmisch-Partenkirchen, Germany
[2]Institute of Geography and Geoecology (IfGG), Karlsruhe Institute of Technology, Kaiserstrasse 12, 76131 Karlsruhe, Germany
[3]Institute of Meteorology and Climatology, Leibniz Universität Hannover, Hannover, Germany
[4]Norwegian Meteorological Institute, Oslo, Norway

*Correspondence to:* matthias.mauder@kit.edu

**Abstract.** Large-eddy simulation (LES) has become a well-established tool in the atmospheric boundary-layer research community to study turbulence. It allows three-dimensional realizations of the turbulent fields, which large-scale models and most experimental studies cannot yield. To resolve the largest eddies in the mixed layer, a moderate grid resolution in the range of 10 to 100 m is often sufficient, and these simulations can be run on a computing cluster with few hundred processors, or even on a workstation for simple configurations. The desired resolution is usually limited by the computational resources. However, to compare with tower measurements of turbulence and exchange fluxes in the surface layer a much higher resolution is required. In spite of the growth in computational power, a high-resolution simulation LES of the surface layer is often not feasible: to fully resolve the energy containing eddies near the surface a grid spacing of O(1 m) is required. One way to tackle this problem is to employ a vertical grid nesting technique, where the surface is simulated at the necessary fine grid resolution, and it is coupled with a standard, coarse, LES that resolves the turbulence in the whole boundary-layer. We modified the LES model PALM (Parallelized Large-eddy simulation Model) and implemented a two-way nesting technique, with coupling in both directions between the coarse and the fine grid. The coupling algorithm has to ensure correct boundary conditions for the fine grid. Our nesting algorithm is realized by modifying the standard third order Runge-Kutta time stepping to allow communication of data between the two grids. The two grids are concurrently advanced in time while ensuring that the sum of resolved and subgrid-scale kinetic energy is conserved. We design a validation test and show that the temporally averaged profiles from the fine grid agree well compared to the reference simulation with high-resolution in the entire domain. The overall performance and scalability of the nesting algorithm is found to be satisfactory. Our nesting results in more than 80 percent savings in computational power for 5 times higher resolution in each direction in the surface layer.

# 1 Introduction

Turbulence in the Atmospheric Boundary Layer (ABL) encompasses a wide range of scales from the boundary-layer scale down to the viscous dissipation scale. In ABL flows, Reynolds numbers (Re) of $10^8$ are commonly encountered. Explicit simulation of the Navier-Stokes equations down to the dissipative scales (DNS: direct numerical simulation) for atmospheric processes is prohibitively expensive, as the required number of grid points in one direction scales with $Re^{3/4}$ (Reynolds, 1990). This corresponds to a three-dimensional ABL simulation domain with total number of grid points of order $10^{17}$. The supercomputers of today cannot fit more than $10^{12}$ grid points in the memory. To be able to compute turbulence processes in the atmosphere nevertheless, the concept of large-eddy simulation (LES) has been introduced already a few decades ago, e.g. Deardorff (1974), Moeng and Wyngaard (1988) and Schmidt and Schumann (1989), where the presence of a subgrid-scheme allows that only the most energetic eddies are resolved.

One of the first large-eddy simulations (LES) by Deardorff (1974) used 64000 grid points to simulate a domain of $5 \text{ km} \times 5 \text{ km} \times 2 \text{ km}$ with a grid resolution of $(125, 125, 50)$ m. The size of one such grid cell is just sufficient to resolve the dominant large eddies and there are just enough grid points to represent the ABL. As computing power progressed, higher resolution and larger domains became possible. By the time of Schmidt and Schumann (1989) the number of grid cells had raised to $160 \times 160 \times 48$, simulating an ABL of $8 \text{ km} \times 8 \text{ km} \times 2.4 \text{ km}$ with a resolution of $(50, 50, 50)$ m. Khanna and Brasseur (1998) used $128^3$ grid points to simulate a domain of $3 \text{ km} \times 3 \text{ km} \times 1 \text{ km}$ to study buoyancy and shear induced local structures of the ABL. Patton et al. (2016) used $(2048, 2048, 1024)$ grid points with a grid resolution of $(2.5, 2.5, 2)$ m to study the influence of atmospheric stability on canopy turbulence. More recently, Kröniger et al. (2018) used $13 \cdot 10^9$ grid points to simulate a domain of $30.72 \text{ km} \times 15.36 \text{ km} \times 2.56 \text{ km}$ to study the influence of stability on the surface–atmosphere exchange and the role of secondary circulations in the energy exchange. The atmospheric boundary-layer community has greatly benefited from the higher spatial resolution available in these LES to study turbulent processes that cannot be obtained in field measurements.

Still, especially in heterogeneous terrain, near topographic elements, buildings or close to the surface the required higher resolution is not always attainable due to computational constraints. In spite of the radical increase in the available computing power over the last decade, large-eddy simulation of high Reynolds number atmospheric flows with very high-resolution in the surface-layer remain a challenge. Considering the size of the domain required to reproduce boundary-layer scale structures, it is computationally demanding to generate a single fixed grid that could resolve all relevant scales satisfactorily. Alternatively, local grid refinement is possible in the Finite-Volume codes that are not restricted to structured grids. Flores et al. (2013) developed a solver for the OpenFOAM modelling framework to simulate atmospheric flows over complex geometries using an unstructured mesh approach. The potential of adaptive mesh refinement technique where the tree-based Cartesian grid is refined or coarsened dynamically, based on the flow structures, is demonstrated by van Hooft et al. (2018). In the Finite-Difference models, a grid nesting technique can be employed to achieve the required resolution. In the nested grid approach, a parent domain with a coarser resolution simulates the entire domain while a nested grid with a higher resolution extends only up to the region of interest. Horizontal nesting has been applied to several mesoscale models (Skamarock et al., 2008; Debreu et al., 2012). Horizontally nested LES-within-LES or LES embedded within a mesoscale simulation is available in the

Weather Research and Forecast model (Moeng et al., 2007). Comparable grid nesting techniques are also widely employed by the engineering turbulence research community but often use different terminology. Nesting in codes with cartesian grids are referred to as local or zonal grid algorithm (Kravchenko et al., 1996; Boersma et al., 1997; Manhart, 2004) and as overset mesh (Nakahashi et al., 2000; Kato et al., 2003; Wang et al., 2014) in unstructured or moving grid codes.

For our purposes, we will focus on vertical nesting, i.e. we consider a Fine Grid nested domain (FG) near the lower boundary of the domain, and a Coarse Grid parent domain (CG) in the entire of the boundary layer. While the latter's resolution ($< 50\,\mathrm{m}$) is sufficient to study processes in the outer region where the dominant eddies are large and inertial effects dominate, such coarse resolution is not sufficient where fine-scale turbulence in the surface layer region is concerned. The higher resolution achieved by the vertical nesting will then allow a more accurate representation of the turbulence in the surface layer region, by

resolving its dominant eddies. For studies that require very high resolution near the surface (e.g. virtual tower measurements, wakes behind obstacles, dispersion within street canyons for large cities) a nesting approach is an attractive solution due to the reduced memory requirement. Challenge of the vertically nested simulation is that the FG upper boundary conditions need to be correctly prescribed by the CG. Though vertical nesting is less common than the horizontal nesting it has been implemented in some LES models. A non-parallelized vertical nesting was explored by Sullivan et al. (1996) but the code is not in public

domain and we could not find any record of further development or application of this code in publications. A LES-within-LES vertical nesting is implemented by Zhou et al. (2018) in the Advanced Regional Prediction System (ARPS) model. We would like to point out that the vertical nesting available in Weather Research and Forecast model (Daniels et al., 2016) is not a conventional vertical nesting because the parent and the child grid still have the same vertical extent, the child grid is only more refined in the vertical.

An analysis of different nesting procedures for mesoscale simulation was performed by Clark and Hall (1991), they coined the terms one-way and two-way interactions. In one-way interaction, only the FG receives information from the CG, and there is no feedback to the CG. In two-way interaction, the FG top boundary conditions are interpolated from the CG and the CG values in the overlapping region are updated with the FG resolved fields. The 'update' process, referred to as 'anterpolation' by Sullivan et al. (1996), is similar to the restriction operation in Multi-Grid methods. Harris and Durran (2010) used a linear

1D shallow-water equation to study the influence of the nesting method on the solution and found the two-way interaction to be superior if the waves are well resolved. They introduce a filtered sponge boundary condition to reduce the amplitude of the reflected wave at the nested grid boundary. We will make use of the interpolation and anterpolation formulas of Clark and Farley (1984). Clark and Hall (1991) studied two different approaches for updating the CG values, namely "post-insertion" and "pressure defect correction". The two approaches were also investigated by Sullivan et al. (1996) in their vertical nesting

implementation. In the post-insertion technique, once the Poisson equation for pressure is solved in the FG, the resolved fields are then anterpolated to the CG. In the pressure defect correction approach, the pressure in the CG and FG are matched by adding a correction term to the CG momentum equations and an anterpolation operation is not required. Though Sullivan et al. (1996) note the pressure defect correction approach to be more elegant, no significant difference in the results was reported.

In the following sections we describe the technical realization and numerical aspects of the two-way nesting algorithm. In

the LES model PALM, a validation simulation is set-up and the results of the nested and standalone simulations are compared.

A second simulation is set-up to evaluate the computational performance of the algorithm. The practical considerations and the limitations of the two-way nesting are then discussed.

## 2 Methods

### 2.1 Description of the standard PALM model

The Parallelized Large-eddy simulation Model (PALM) is developed and maintained at the Leibniz University of Hannover (Raasch and Schröter, 2001; Maronga et al., 2015). We give a quick summary of the model here and highlight the aspects which will reappear when discussing our nesting modifications. PALM is a finite difference solver for the non-hydrostatic incompressible Navier-Stokes equations in the Boussinesq approximation. PALM solves for six prognostic equations: the three components of the velocity field $(u, v, w)$, potential temperature $(\theta)$, humidity $(q)$ and the sub-grid scale kinetic energy $(e)$.
The sub-grid scale (SGS) turbulence is modelled based on the method proposed by Deardorff (1980). The equations for the conservation of mass, energy and moisture (Eqs. 1, 2, 3 and 4) are filtered over a grid volume on a Cartesian grid. Adopting the convention of Maronga et al. (2015), the overbar denoting the filtered variables are omitted. However, the overbar is shown for SGS fluxes. The SGS variables are denoted by a double prime. The prognostic equations for the resolved variables are:

$$\frac{\partial u_i}{\partial t} = -\frac{\partial u_i u_j}{\partial x_j} - \varepsilon_{ijk} f_j u_k + \varepsilon_{i3k} f_3 u_{k_{g,j}} - \frac{1}{\rho_0} \frac{\partial \pi^*}{\partial x_i} + g \frac{\theta_v - \langle \theta_v \rangle}{\theta_v} \delta_{i3} - \frac{\partial}{\partial x_j} \left( \overline{u_i'' u_j''} - \frac{2}{3} e \delta_{ij} \right), \tag{1}$$

$$\frac{\partial u_j}{\partial x_j} = 0, \tag{2}$$

$$\frac{\partial \theta}{\partial t} = -\frac{\partial u_j \theta}{\partial x_j} - \frac{\partial}{\partial x_k} \left( \overline{u_j'' \theta''} \right) - \frac{L_V}{c_p \Pi} \Psi_{q_v}, \tag{3}$$

$$\frac{\partial q_v}{\partial t} = -\frac{\partial u_j q_v}{\partial x_j} - \frac{\partial}{\partial x_k} \left( \overline{u_j'' q_v''} \right) + \Psi_{q_v}. \tag{4}$$

The symbols used in the above equations are listed in Table 1. The 1.5 order closure parameterization modified by Moeng and Wyngaard (1988) and Saiki et al. (2000), assumes a gradient diffusion parameterization (Eqs. 6, 7, 8). The prognostic equation for the SGS-TKE reads as

$$\frac{\partial e}{\partial t} = -u_j \frac{\partial e}{\partial x_j} - \left( \overline{u_i'' u_j''} \right) \frac{\partial u_i}{\partial x_k} + \frac{g}{\theta_{v,0}} \overline{u_3'' \theta_v''} - 2K_m \frac{\partial e}{\partial x_j} - \epsilon, \tag{5}$$

with the SGS fluxes modelled as:

$$\overline{u_i'' u_j''} - \frac{2}{3} e \delta_{ij} = -K_m \left( \frac{\partial u_i}{\partial x_j} + \frac{\partial u_j}{\partial x_i} \right), \tag{6}$$

$$\overline{u_i'' \theta''} = -K_h \frac{\partial \theta}{\partial x_i}, \tag{7}$$

and

$$\overline{u_i'' q_v''} = -K_h \frac{\partial \theta}{\partial x_i}. \tag{8}$$

The eddy diffusivities are proportional to $e^{3/2}$ under convective conditions (Maronga et al., 2015). For a thorough description of the governing equations and parameterizations, see Maronga et al. (2015).

**Table 1.** List of symbols in the governing equations and parameterizations.

| Symbol | Description |
|---|---|
| $f_i$ | Coriolis parameter |
| $\rho_0$ | Density of dry air at the surface |
| $\pi^*$ | Modified perturbation pressure |
| g | Gravitational acceleration |
| $\theta_v$ | Virtual potential temperature |
| $L_v$ | Latent heat of vaporization |
| $C_p$ | Heat capacity of dry air at constant pressure |
| $q_v$ | Specific humidity |
| $\Psi_{q_v}$ | Source/sink term of $q_v$ |
| $\Pi$ | Exner function for converting between temperature and potential temperature |
| $K_h$ | SGS eddy diffusivity of heat |
| $K_m$ | SGS eddy diffusivity of momentum |

The prognostic equations are discretized on a staggered Arakawa C-grid, where the scalars are evaluated in the center of the grid volume and velocities are evaluated at the center of the faces of the grid volume in their respective direction. The advection terms are evaluated either with fifth-order upwind discretization according to Wicker and Skamarock (2002) or with a 2nd order scheme according to Piacsek and Williams (1970). The prognostic equations are integrated in time using a third-order Runge-Kutta (RK3) scheme. The low storage RK3 scheme with three sub-steps proposed by Williamson (1980) guarantees a stable numerical solution. The Poison Equation for pressure is solved with Fast-Fourier Transform (FFT) when periodic boundary conditions are applied in the lateral boundaries. There are three FFT algorithms available in PALM with FFTW being the optimal method for large scale simulations. Monin-Obukhov Similarity Theory (MOST) is assumed between the surface and the first grid point. A vertical zero pressure gradient at the surface guarantees the vertical velocity to be zero. Simulations can be driven by either prescribing the surface temperature or the surface sensible heat flux, similarly for the humidity. At the top

of the simulation domain the horizontal velocities equal geostrophic wind and the vertical velocity is set to zero. The pressure can assume either a Dirichlet condition of zero value or a Neumann condition of zero vertical gradient. The scalar values can have either a fixed value Dirichlet condition or a fixed gradient Neumann condition. The vertical gradient of SGS Turbulent Kinetic Energy (TKE) is set to zero at both top and bottom boundaries.

5 PALM is a parallelized model and the standard way of parallelization is by dividing the three-dimensional domain into vertical columns, each of which is assigned to one processing element (PE). Each vertical column possesses a number of ghost points needed for computation of derivatives at the boundary of the sub-domains. Each PE can only access data for a single sub-domain. All PEs execute the same program on a different set of data. For optimum load balancing between the PE the decomposed sub-domains should have the same size. In PALM, this condition is always satisfied as only sub-domains of 10 the same size are allowed. The data exchange between PEs needed by the Poisson solver and to update the ghost points are performed via the Message Passing Interface (MPI) communication routines.

## 2.2 Nested model structure

### 2.2.1 Fine grid and coarse grid configuration

We are interested in achieving an increased resolution only in the surface-layer, the lowest 10% of the boundary layer, where 15 surface exchange processes occur and where eddies generated by surface heterogeneity and friction are smaller than the dominant eddies in the mixed layer. We setup the LES-within-LES case by maintaining the same horizontal extent for the FG and the CG to have the whole surface better resolved. We allow the vertical extent of the FG to be varied as needed, typically up to the SL depth. This implementation of vertical grid nesting has two main challenges. The first challenge, that is purely technical in nature, is to implement routines that handle the communication of data between the CG and the FG. The second and the 20 most important challenge is to ensure that the nesting algorithm yields an accurate solution in both grids.

Below we use upper case symbols for fields and variables in the CG, and lower case for the FG. E.g. $E$ and $e$ denote the subgrid-scale turbulent kinetic energy (a prognostic variable in our LES) of CG and FG respectively. The nesting ratio is defined as the ratio of the CG spacing to the FG spacing, and $n_x = \Delta X/\Delta x$, corresponding symbols apply for $y$ and $z$ directions. The nesting ratios $n_x$, $n_y$ and $n_z$ have to be integer. It is possible to have either odd or even nesting ratio and it can be different in 25 each direction. As the domain that is simulated in the FG is completely inside of the CG domain, each FG cell belongs to a CG cell. The two grids are positioned in such a way that a FG cell belongs to only one CG cell and one CG cell is made up by a number of FG cells given by the product of the nesting ratios $n_x \times n_y \times n_z$. This means that if the grid nesting ratio is odd, there will be one FG cell whose center is exactly at the same position as the center of the coarse cell as shown in Fig. 1 (b). The collection of FG cells that correspond to one CG cell is denoted by $\mathcal{C}(I, J, K)$, the collection of FG faces that corresponds 30 to e.g. an yz-face of the CG is denoted by $\mathcal{C}_x(I_s, J, K)$, where it is understood that the $I_s$ index is an index on the staggered grid in the x-direction to denote the position of the face, and similar for the other types of faces. We have used $f_x = 1/n_x$ to denote the inverse of the nesting ratio in the $x$ dimension (corresponding symbols for $y$ and $z$). A schematic diagram of the overlapping grids is shown in Fig. 1 (a).

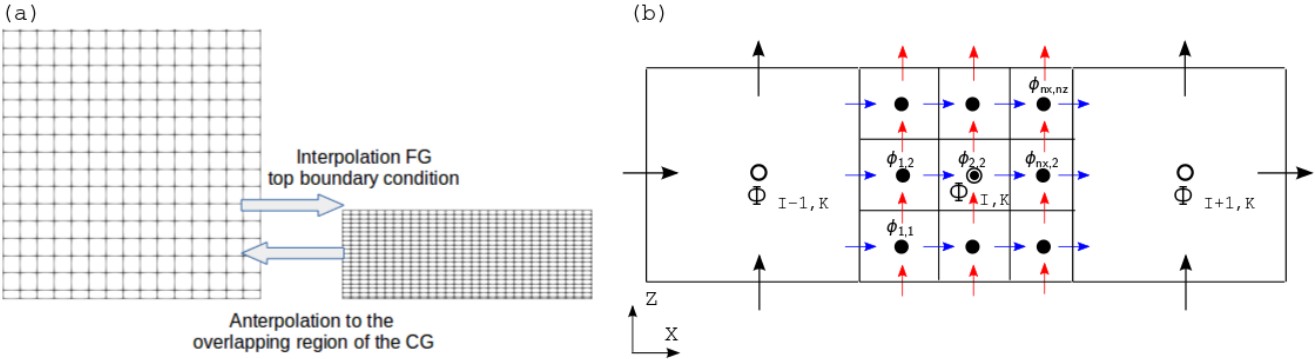

**Figure 1.** (a) Schematic of the interpolation and anterpolation between grids. The FG top boundary condition is interpolated from the CG. The CG prognostic quantities in the overlapping region are anterpolated from the FG. (b) Schematic of Arakawa C grid for two grids with nesting ration of three. The black arrows and circles are CG velocity and pressure, respectively. The blue and red arrows are horizontal and vertical velocity, respectively, in the FG. The filled black circle is the FG pressure. The symbols $\Phi$ and $\phi$ represent CG and FG scalar quantities. Where I and K are CG indices and nx and nz are the nesting ratio in x and z, respectively.

### 2.2.2 Vertical nesting algorithm

We implement a two-way interaction algorithm, shown in Fig. 2, because in our first trials we found that one-way nesting did not improve the FG representation satisfactorily and hence was not pursued further. The FG prognostic quantities are initialized by interpolating the CG values in the overlapping region. Optionally, the initialization of the FG can be delayed until the CG

has reached a fully turbulent state. Both grids are restricted to have identical time steps. PALM finds the largest time step for each grid such that the CFL condition is individually satisfied and the minimum of the two values is then chosen as the time integration step for both grids. The right hand side of the prognostic equation except for the pressure is first computed concurrently in both grids. The values of $u, v, w, \theta$ and $q$ are then anterpolated to the CG in the overlapping region. The CG solves a Poisson equation for pressure. The new $u, v, w, \theta$ and $q$ fields in the CG are interpolated to set the FG Dirichlet top

boundary conditions. The Poisson equation is then solved for pressure in the FG and the vertical velocity in the FG is also updated by the pressure solver at this stage. Since all the velocity components follow Dirichlet condition at FG top boundary only Neumann condition is suitable for pressure (Manhart, 2004). PALM permits the use of a Neumann zero-gradient condition for pressure at both top and bottom boundary. It is advisable to use a Neumann boundary condition at the top and the bottom for the CG too. The TKE is then anterpolated maintaining the Germano identity and it is followed by the computation of SGS

eddy diffusivity for heat ($k_h$) and momentum ($k_m$) in the CG. This procedure is repeated at every sub-step of the Runge-Kutta 3 time integration and it ensures that the velocity field remains divergence free in both grids.

    In the 1.5 order turbulence closure parameterization all the sub-grid fluxes are derived from the turbulent kinetic energy and the resolved gradients at each time step. Therefore, the sub-grid fluxes do not have to be interpolated from CG to FG at the top boundary. Furthermore, in our implementation of the nesting method, we assume that most of the TKE is resolved well

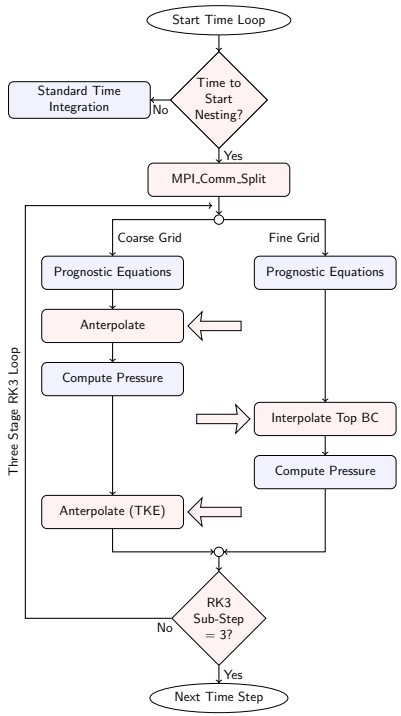

**Figure 2.** A flowchart of the two-way interaction algorithm. The new routines needed for the vertical nesting are highlighted in red and the standard routines are highlighted in blue. An arrow pointing to the left indicates transfer of data from FG to CG, and vice versa.

down to the inertial subrange, except for the lowest few grid layers. This allows us to use the zero-gradient Neumann boundary condition for TKE at the FG top boundary. We employ a simplified sponge layer by limiting the anterpolation of all prognostic quantities to one CG cell less than the nested height. This segregation of the anterpolation region in the CG and top boundary condition level of the FG ensures that the flow structures in the CG propogate into the FG without distortion due to numerical 5 artifacts.

## 2.3 Translation between grids

### 2.3.1 Interpolation

For the boundary conditions at the top of the FG, the fields from the CG are interpolated to the FG, according to Clark and Farley (1984). We define the top of the FG as the boundary level just above the prognostic level of each quantity. In Eq. 10, $\Phi$ 10 and $\phi$ represent CG and FG quantities, respectively. For the scalar fields, the interpolation is quadratic in all three directions. For the velocity components, the interpolation is linear in its own dimension, and quadratic in the other two directions. The same interpolation formulation is also used to initialize all vertical levels of the fine grid domain at the beginning of the nested

simulation. The interpolation is reversible as it satisfies the conservation condition of Kurihara et al. (1979):

$$< \phi >=< \Phi > . \tag{9}$$

For clarity, we illustrate the interpolation by focusing on one particular dimension, in this case $x$, but the same operation holds for $y$ and $z$. The interpolation in the $x$-dimension reads as

$$\phi_m = \eta_-^m \Phi_{I-1} + \eta_0^m \Phi_I + \eta_+^m \Phi_{I+1}, \tag{10}$$

with $m$ running from 1 to $n_x$, thus producing $n_x$ equations for each CG cell $I$. For the interpolation in $y$ and $z$ there will be two additional indices, producing $n_x \times n_y \times n_z$ equations for all the FG cells corresponding to the CG parent cell. For the quadratic interpolation a stencil with 3 legs is used, relating the prognostic value of a FG cell to the value of its parent CG and the values of the immediate CG neighbour on the left and on the right of the parent cell, e.g. $\Phi_{I-1}$ and $\Phi_{I+1}$ for the x direction as shown

in Fig. 1 (b). The stencil coefficients are:

$$\eta_-^m = \frac{1}{2} H_m (H_m - 1) + \alpha,$$
$$\eta_0^m = (1 - H_m^2) - 2\alpha, \tag{11}$$
$$\eta_+^m = \frac{1}{2} H_m (H_m + 1) + \alpha,$$

with the weights $H_m$ expressed in function of the inverse nesting ratio,

$$H_m = \frac{1}{2}((2m-1)f_x - 1), \tag{12}$$

and the coefficient $\alpha$ is chosen such that the conservation condition of Kurihara et al. (1979) is satisfied,

$$\alpha = \frac{1}{24} \left( f_x^2 - 1 \right). \tag{13}$$

It can be observed that the sum of the $\eta$'s equals 1.

### 2.3.2 Anterpolation

The anterpolation of the prognostic quantities are performed by an averaging procedure according to Clark and Hall (1991). The anterpolation equations for the velocities read as:

$$U_{I,J,K} =< u >_{j,k}= \sum_{j,k \in \mathcal{C}_{IJK}} u_{i^*,j,k} f_y f_z,$$

$$V_{I,J,K} =< v >_{i,k}= \sum_{i,k \in \mathcal{C}_{IJK}} v_{i,j^*,k} f_x f_z, \tag{14}$$

$$W_{I,J,K} =< w >_{i,j}= \sum_{i,j \in \mathcal{C}_{IJK}} w_{i,j,k^*} f_x f_y.$$

For the scalars it is:

$$\Phi_{I,J,K} = [\phi]_{i,j,k} = \sum_{i,j,k \in \mathcal{C}_{IJK}} \phi_{i,j,k} f_x f_y f_z. \tag{15}$$

Here the lower case indices only count over the fine grid cells that belong to that particular coarse grid cell. For each $(I, J, K)$ tuple of a parent CG cell there exists a set $\mathcal{C}_{\mathcal{IJK}}$ containing the $(i, j, k)$ tuples of its corresponding children FG cells. To ensure that the nested PALM knows at all times which fine grid cells and coarse grid cells correspond, we compute this mapping for the FG and CG indices before starting the simulation, and we store it in the memory of the parallel processing element. In the Arakawa C-grid discretization that PALM uses, the scalars are defined as the spatial average over the whole grid cell, and therefore it is required that the CG scalar is the average of the corresponding FG scalars in (Eq. 15). However, the velocities are defined at the faces of the cells in the corresponding dimension. Therefore in (Eq. 14) the CG velocity components are computed as the average over the FG values at the FG cells that correspond to the face of the CG cell, expressed by $i^*$, $j^*$, $k^*$ respectively.

However, the TKE in the CG differs from the FG value, due to the different resolution of grids. In the FG the SGS motions are weaker because the turbulence is better resolved. Therefore, TKE is anterpolated such that the sum of resolved kinetic energy and TKE (SGS kinetic energy) is preserved, by maintaining the Germano identity (Germano et al., 1991):

$$E = [e] + \frac{1}{2} \sum_{n=1}^{3} \left( [u_n u_n] - [u_n][u_n] \right). \tag{16}$$

Here the straight brackets are the spatial average over the coarse grid cell ($f_x f_y f_z \times \sum_{i,j,k \in \mathcal{C}_{IJK}}$) and the $n$ index runs over the three spatial dimensions. In other words, to obtain the CG TKE from the average FG TKE, we add the variance of the FG velocity components over the FG cells comprising the CG cell. Therefore CG TKE is always larger than FG TKE.

## 2.4 Parallel Inter-Grid Communication

MPI is the most widely used large scale parallelization library. The atmosphere-ocean coupling in PALM has been implemented following MPI-1 standards (Esau, 2014; Maronga et al., 2015). We follow a similar approach for the MPI communications, and have adopted MPI-1 standards for our nesting implementation. Concurrent execution of the two grids is achieved with the MPI_COMM_SPLIT procedure. The total available processors are split into two groups, denoted by color 0 or 1 for CG and FG respectively, see Fig. 3. The data between the processors of the same group are exchanged via the local communicator created during the splitting process, whereas the data between the two groups are exchanged via the global communicator MPI_COMM_WORLD.

Based on the nesting ratio and the processor topology of the FG and the CG group a mapping list is created and stored. Given the local PE's 2D processor co-ordinate – the list will identify the PEs in the remote group to/from which data needs to be sent/received; the actual communication then takes place via the global communicator. There are three types of communication in the nesting scheme:

i. Initializing the FG (Send data from coarse grid to fine grid.) This is performed only once.

ii. Boundary condition for the FG top face (Send data from coarse grid to fine grid.).

iii. Anterpolation (Send data from fine grid to coarse grid.).

The exchange of arrays via MPI_SENDRECV routines is computationally expensive. Therefore, the size of the arrays communicated are minimized by performing the anterpolation operation in the FG PE's and storing the values in a temporary 3D

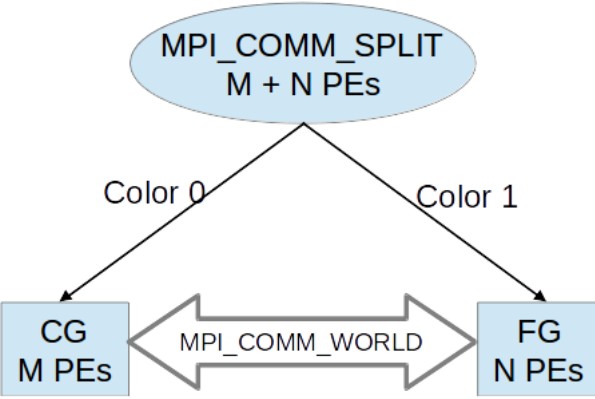

**Figure 3.** Schematic of the MPI processor grouping. The data exchange between the two groups are performed via the global communicator. M and N are the number of processors for CG and FG respectively.

array that is later sent via the global communicator to the appropriate CG PE. This approach is more efficient than performing the anterpolation operation on the CG which has less PE's and needs communication of larger arrays from the FG. Furthermore, the array data that need to be communicated during the anterpolation operation and for setting the FG boundary condition are not contiguous in memory. The communication performance is enhanced by creating an MPI derived data type that ensures
that the data is sent contiguously. Within the RK3 sub-steps, when one grid executes the pressure solver, the other grid has to wait, leading to more computational time at every sub-step. However, the waiting time can be minimized by effective load balancing, i.e. the number of grid points per PE in the CG should be kept lower than in the FG. The reduction in workload per CG PE is achieved with a few additional cores. The reduction in computational time per step in the CG means the waiting time on the FG PE is also reduced.

**3    Results and Discussion**

**3.1    Simulation setup for the nesting validation test**

To evaluate the accuracy of the two-way nesting algorithm we setup a convective boundary layer simulation. Two overlapping grids with a nesting ratio of five in the lateral and vertical direction are employed. The simulation parameters are listed in Table 2. A standalone reference simulation with the same resolution as the coarse grid (SA-C) and another reference with the
same resolution as the fine grid (SA-F) are performed for comparison. The grid configuration and the computational resources used are listed in Table 3. The simulations were performed in a local computing cluster, each compute node has 64 GB of main memory and a 2.8 GHz Ivy Bridge processor with 20 cores. The simulation domain has periodic boundary conditions in the

lateral direction. The Dirichlet boundary condition is applied for velocity at the top and bottom boundaries, the vertical velocity component is set to zero and the horizontal components are set to geostrophic wind. At the top and bottom boundaries, the pressure and humidity are set to zero gradient Neumann condition. The potential temperature is set to a Neumann condition at the bottom, and the gradient is determined by MOST based on the prescribed surface heat flux and roughness length. The gradient of the initial profile is maintained at the top boundary. In PALM, $u_g$ and $v_g$ represents the $u$- and $v$-component of the geostrophic wind at the surface. The $u$ and $v$ initial profiles are set to be constant, equal to the value of the geostrophic wind component in the domain and the vertical velocity is initialized to zero in the domain. The potential temperature is initialized to a constant value of 300 K up to 800 m and above 800 m a lapse rate of $1\,\mathrm{K}\,(100\,\mathrm{m})^{-1}$ is prescribed. The humidity profile is initialized to a constant value of $0.005\,\mathrm{kg\,kg^{-1}}$. The simulation is driven by prescribing a surface heat flux of $0.1\,\mathrm{K\,m\,s^{-1}}$ and a surface humidity flux of $4 \times 10^{-4}\,\mathrm{kg\,kg^{-1}m\,s^{-1}}$. The domain is more than four times larger in the horizontal than the initial boundary layer height.

**Table 2.** Simulation Parameters for the nesting validation test.

| Simulation Parameters | Value |
| --- | --- |
| Domain Size: | 4.0 x 4.0 x 1.65 km$^3$ |
| Fine grid vertical extent: | 320 m |
| Kinematic surface heat flux: | $H_s = 0.1\,\mathrm{K\,m\,s^{-1}}$ |
| Kinematic surface humidity flux: | $\lambda E_s = 4 \times 10^{-4}\,\mathrm{kg\,kg^{-1}m\,s^{-1}}$ |
| Geostrophic wind: | $u_g = 1\,\mathrm{m\,s^{-1}}$ , $v_g = 0\,\mathrm{m\,s^{-1}}$ |
| Roughness length | 0.1 m |
| Simulated time: | 10800 s |
| Spin-up time: | 9000 s |
| Averaging interval: | 1800 s |

### 3.2 Analysis of the simulations

In a two-way nesting it is important that the flow structures are propagated from the FG to CG and vice versa, without any distortion. In Fig. 4, the contours in the CG region overlapping the FG have similar structures as the FG. The higher resolution in the FG enables more detailed contours whereas the anterpolated CG contours are smoother. Furthermore, in the CG region beyond the overlapping region no distortion to the contours are observed indicating that the anterpolation does not introduce sharp gradients in the CG.

**Table 3.** Grid configuration of the nested and standalone reference domains.

| Case | No. of grid points | (dx,dy,dz) m | CPU cores | Core-hours | Grid points per core | Time steps |
|---|---|---|---|---|---|---|
| Coarse Grid (CG) | 200 x 200 x 80 = 3.2 x $10^6$ | 20, 20, 20 | 20 | 376 | 1.6 x $10^5$ | 17136 |
| Fine Grid (FG) | 1000 x 1000 x 80 = 80 x $10^6$ | 4, 4, 4 | 80 | 1503 | 1.0 x $10^6$ | 17136 |
| Total | | | | 1879 | | |
| Standalone Coarse (SA-C) | 200 x 200 x 80 = 3.2 x $10^6$ | 20, 20, 20 | 20 | 8 | 1.6 x $10^5$ | 3226 |
| Standalone Fine (SA-F) | 1000 x 1000 x 400 = 400 x $10^6$ | 4, 4, 4 | 400 | 8234 | 1.0 x $10^6$ | 18343 |

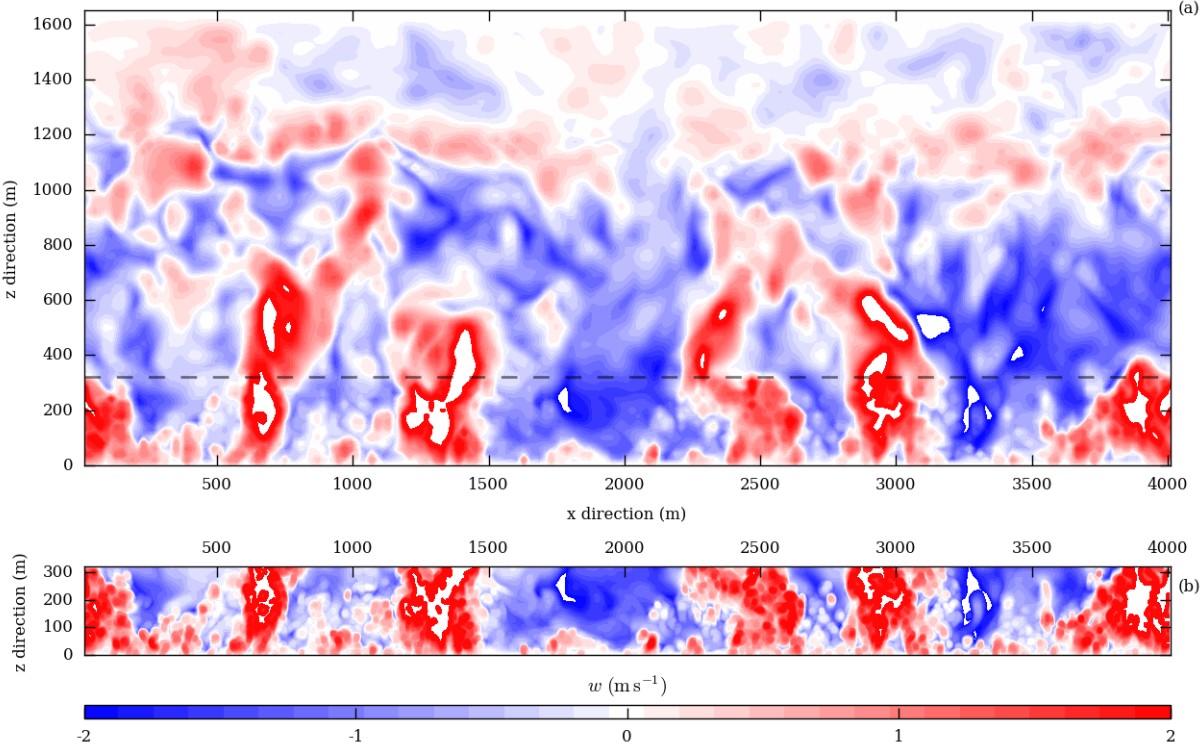

**Figure 4.** Instantaneous contours of vertical velocity, (a) CG and (b) FG, at the vertical x-z cross-section at the center of the domain after 10800 s of the simulation. The dashed line in (a) marks the top of the overlapping region. Flow structures in the FG, are similar but more detailed than the CG, qualitatively indicate the improvement to the surface-layer resolution with the two-way nesting.

Vertical profiles are used for quantitative comparison of the nested and the reference simulations. The turbulent fluctuations (e.g. $\theta''$,$w''$) are defined as the spatial deviations from the instantaneous horizontal average. The turbulent fluxes (e.g. $< \overline{w''\theta''} >, < \overline{u''u''} >$) are obtained using the spatial covariance and are then horizontally averaged. All the horizontally averaged profiles (e.g. $< \theta >, < w''\theta'' >$) are also averaged over time but we omit the conventional overline notation for readability. The

convective velocity scale ($w_*$) and temperature scale ($\theta_*$) obtained from SA-F are used to normalize the profiles. The convective velocity is calculated as $w_* = (g\,\theta_0^{-1}\,H_s\,z_i)^{1/3}$, where g is the gravitational acceleration, $\theta_0$ is the surface temperature and $z_i$ is the boundary layer height in the simulation. The convective temperature scale is calculated as $\theta_* = H_s\,w_*^{-1}$. In Fig. 5 (a and c), the vertical profiles of difference between the potential temperature ($<\theta>$) and its surface value normalized by the convective temperature scale are plotted. Since the FG profiles are superior to the CG in the overlapping region, the anterpolated CG values are not plotted. In Fig. 5 (a), there is no visible difference between the standalone and the nested simulations. However, in the region closer to the surface, plotted in Fig. 5 (c), a better agreement between the SA-F and FG is observed. The potential temperature variance ($<\theta''\theta''>$) normalized by the square of the temperature scale ($\theta_*^2$) is shown in Fig. 5 (b and d). Here too FG provides better accuracy close to the surface.

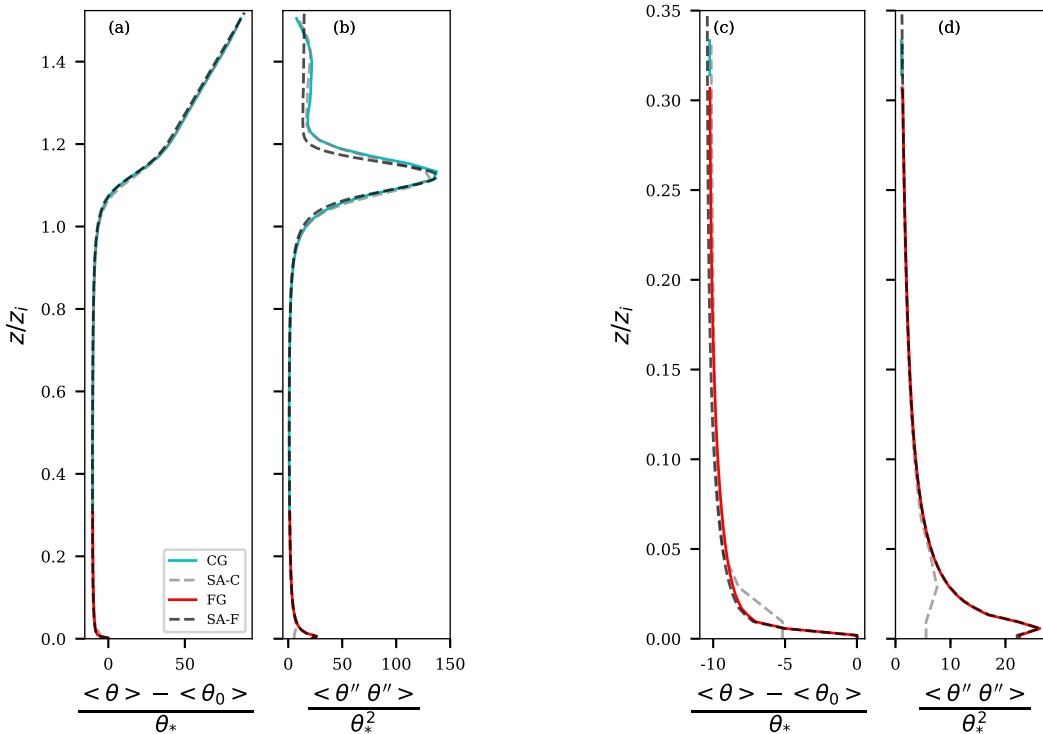

**Figure 5.** Vertical profile of horizontally averaged potential temperature normalized by surface value (a and c) and variance of potential temperature normalized by $\theta_*^2$ (b and d). The nested grid profiles agree well with the SA-F in the surface layer. The improvement of the two-way nesting, at the boundary layer height, is seen in the good agreement in the profiles of CG and SA-F in (b).

In the vertical heat flux ($<w''\theta''>$) profiles in Fig. 6, the FG has good agreement with the SA-F in the surface layer for the resolved, SGS and the total flux profiles. In the CG regions above the nested grid height, a good agreement with the SA-C is found as well. The improvement due to the two-way nesting is seen in Fig. 6 (d and e), where the effects of low grid resolution

of the SA-C in resolved and SGS fluxes are evident. However, no grid dependent difference in the profile is observed in the total flux.

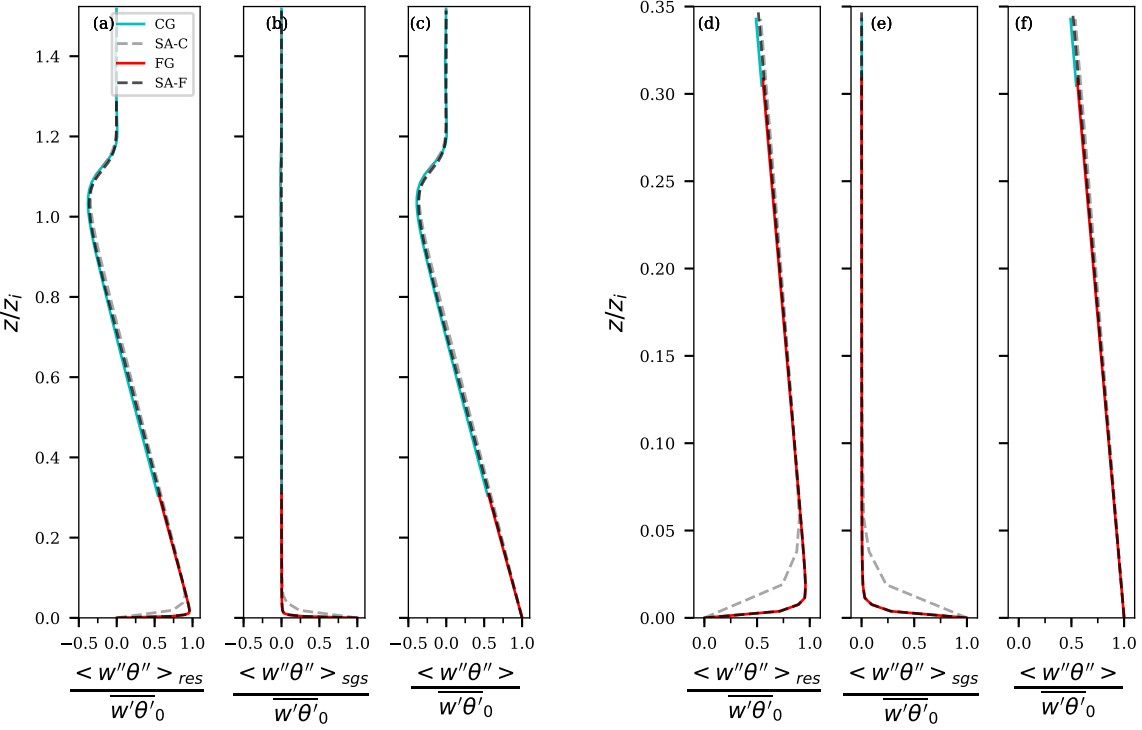

**Figure 6.** Vertical profile of horizontally averaged heat flux normalized by the surface heat flux – resolved (a and d), sub-grid (b and e), and total flux (c and f). The two-way nesting significantly improves the resolved and SGS fluxes in the surface layer.

The resolved variances of $u$, $v$ and $w$ normalized by the square of the convective velocity ($w_*^2$) are plotted in Fig. 7. The FG $v$ and $w$ FG profiles have a better agreement with the SA-F than the $u$ variance. The $u$ and $v$ variances in Fig. 7 (d and e) lie between SA-C and SA-F indicating that the resolved variances are improved compared to the SA-C but not sufficiently resolved to match SA-F. At the nesting height the variances deviate more from the SA-F and approach the CG values. Due to conservation of total kinetic energy across the nest boundary, more CG TKE is contained in the sub-grid scale. Consequently, the resolved CG variances could have an undershoot as compared to SA-F, resulting in an undershoot of the FG variances too at the nesting height. Above the nesting height, the variance of $u$, $v$ and $w$ in CG are similar to SA-C.

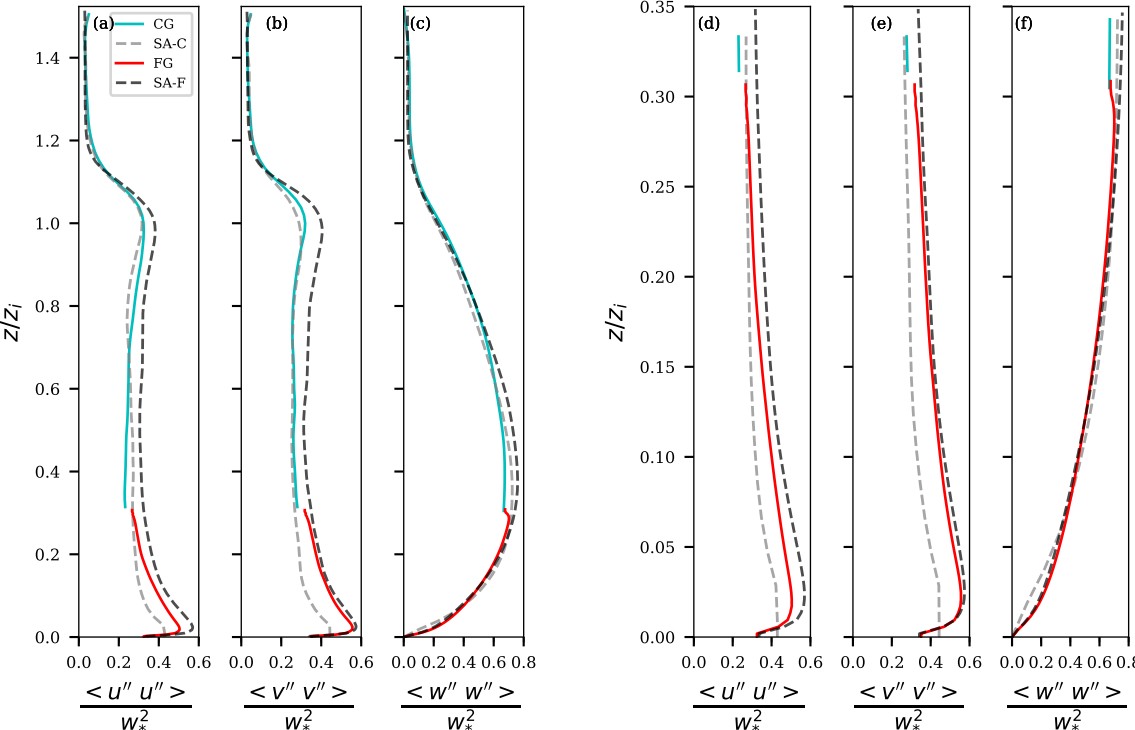

**Figure 7.** Vertical profile of horizontally averaged resolved variance of $u$ (a and d), $v$ (b and e), and $w$ (c and f) normalized by $w_*^2$. The variance of $v$ and $w$ show better agreement with the standalone reference in the surface layer.

The resolved vertical velocity skewness in Fig. 8 shows good agreement between the FG and SA-F close to the surface. However, at the nesting height a small kink in the skewness is noticeable. Zhou et al. (2018) observe that the magnitude of the kink in the higher-order profiles can be minimized by increasing the depth of the sponge layer. Our simplified sponge layer approach appears to be unable to effectively minimize the kinks at the nesting height. The resolved skewness in CG is lower than SA-C possibly due to larger SGS TKE in the CG, as seen in Fig. 8 (d). The SGS TKE in Fig. 8 (d) shows exact match between FG and SA-F close to the surface and only marginal difference at the nesting height. However, CG values are considerably different from the SA-C values close to the surface due to the anterpolation maintaining Germano identity for conservation of kinetic energy across the grids. In the coarse resolution SA-C, near the surface, the SGS turbulence model appears to insufficiently model the SGS effects. Above the nesting height the CG is similar to SA-C.

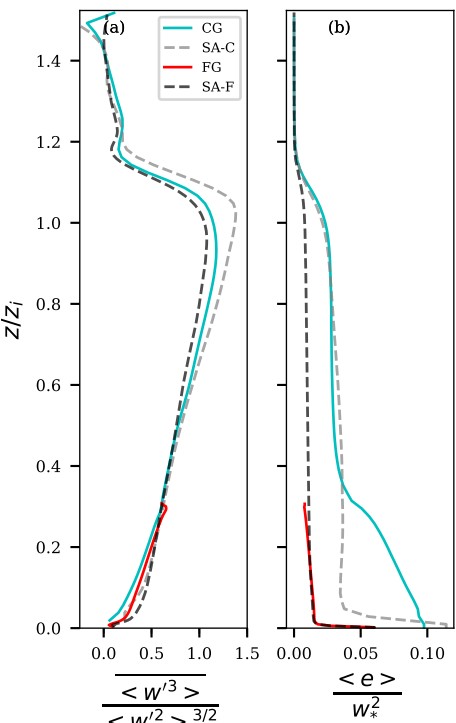 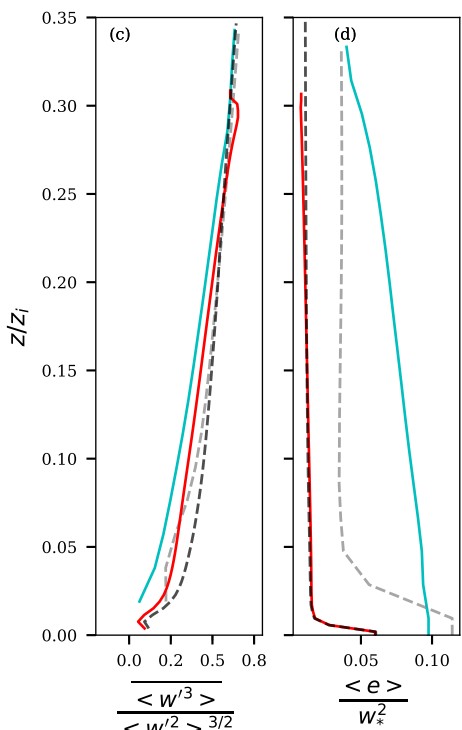

**Figure 8.** Vertical profile of horizontally averaged resolved vertical velocity skewness (a and c), and SGS turbulent kinetic energy $e$ (b and d) normalized by $w_*^2$. The SGS TKE in the CG is higher than SA-C as a result of anterpolation maintaining the Germano identity.

The horizontal spectra of SGS turbulent kinetic energy and vertical velocity are plotted in Fig. 9 at two levels, one within the nested grid and one above the nested grid height. The FG TKE spectra in Fig. 9 (c) perfectly overlaps the SA-F spectra. The CG spectra has higher energy than the SA-C, this corresponds to the higher CG TKE values observed in Fig. 8 (c). As the limit of the grid resolution is reached at high wavenumber, the drop in the CG spectra is marginally shifted compared to SA-C. This improvement at high wavenumber is due to feedback from the FG. Similarly, the vertical velocity spectra in Fig. 8 (d) shows marginal improvement at high wavenumber for the CG with respect to SA-C. While the FG agrees with SA-C at high wavenumber and at the spectra peak, at low wavenumber FG follows the CG spectra. At the level above the nested grid, the CG spectra agrees with SA-C for both TKE and the vertical velocity.

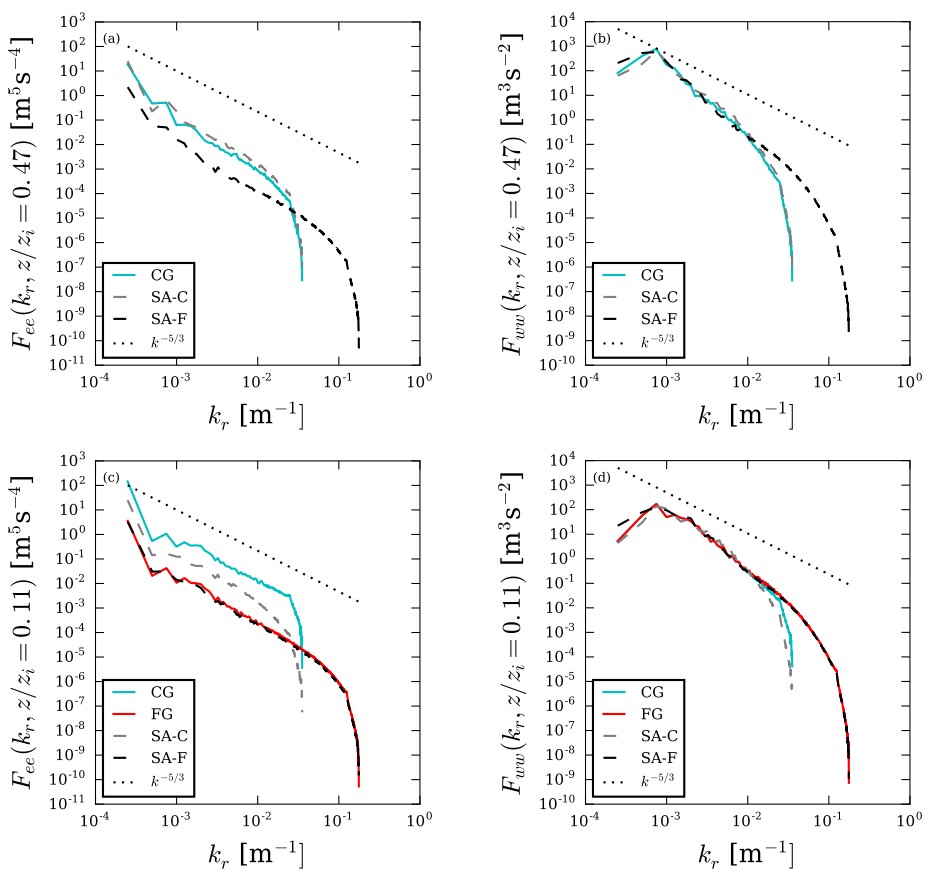

**Figure 9.** Spectra of SGS turbulent kinetic energy ($e$) (a and c), and vertical velocity ($w$) (b and d). At $z/z_i$ = 0.47 (a and b) and at $z/z_i$ = 0.11 (c and d); $k_r$ is the horizontal wavenumber.

## 3.3 Computational Performance

The computational resources used in the simulations discussed above are listed in Table 3. The resources needed by SA-C is only 8 core hours. While the nested simulations needed about 1879 core hours, the SA-F needed about 4 times more core hours than the nested simulation. As the resolution is increased from $20\,\mathrm{m}$ in SA-C to $4\,\mathrm{m}$ in SA-F, the number of time steps increased more than 5 times as higher resolution demands smaller time step size. Though the number of time steps in FG is similar to SA-F, limiting the nested grid in the vertical direction has reduced the number of CPU cores needed, and higher resolution in the surface layer is achieved at a reduced computational cost.

Several factors influence the computational performance of an LES code. Some factors depend on the hardware, for e.g. the number of grid points per PE depends on the memory available per node. On the other hand, the communication time for data exchange between the PEs depend on the topology of the domain decomposition. The best performance in terms of

**Table 4.** Number of grid points in nested and non-nested FG domain.

| Case | No. of grid points |
|---|---|
| Coarse Grid | 840 x 840 x 288 = 0.20 x $10^9$ |
| Fine Grid | 4200 x 4200 x 360 = 6.35 x $10^9$ |
| Total | 6.55 x $10^9$ |
| Non-nested FG | 4200 x 4200 x 360 = 6.35 x $10^9$ |

**Table 5.** Grid configuration of the nested and non-nested FG domain.

| | Nested | | | | | Non-Nested FG | | |
|---|---|---|---|---|---|---|---|---|
| Run | Total PE | CG PE | FG PE | Avg. time per step [s] | Efficiency [%] | Total PE | Avg. time per step [s] | Efficiency [%] |
| A | 1664 | 64 | 1600 | 44.0 | 100 | 1600 | 14.9 | 100 |
| B | 3744 | 144 | 3600 | 19.9 | 98 | 3600 | 6.7 | 99 |
| C | 7488 | 288 | 7200 | 10.3 | 95 | 7200 | 3.6 | 92 |
| D | 8736 | 336 | 8400 | 9.3 | 90 | 8400 | 3.4 | 84 |
| E | 14976 | 576 | 14400 | 5.6 | 87 | 14400 | 2.3 | 74 |

communication time in a standalone run is achieved when the number of sub-domains in the x and y directions are equal. In that case the number of ghost points at the lateral boundaries are optimally minimized. In a nested simulation, the load per PE, i.e. the number of grid points per PE, in the two grids varies. As the speed of the model integration depends on the PE load, the load balancing between fine and coarse grid has an effect on the computational performance of the nested simulation. Keeping these factors in mind, we designed the nested simulation domains listed in Table 4 for the purpose of assessing the computational performance, as the total number of processors is varied. To avoid load balancing bias in the scalability analysis, the ratio between the number of PEs for CG and FG is kept constant in all the five runs listed in Table 5. Keeping the processor ratio constant implies that the ratio between the number of grid points per PE in CG and FG is also held constant. Consequently, in this performance test, the FG has 1.25 times more grid points per PE than the CG in all the processor configurations tested. To compare the performance of nested model against the non-nested version of PALM under equivalent work load, a grid with the same dimensions of the FG is set-up. This non-nested grid also has the same load per PE and same number of cores as the FG. Such a non-nested set-up is acceptable for comparison since the number of PE in CG is negligible compared to the PE in FG in our set-up (e.g. 14400 PE in FG and only 576 PE in CG). A pure standalone simulation with FG resolution throughout the boundary layer was not performed as it would need about 2.5 x $10^{10}$ grid points and such a large domain was computationally not feasible.

The performance is measured in terms of the time taken to simulate one time step. To increase the accuracy of this performance measurement, the simulation is integrated for ten time steps and the average of the time per step is plotted. The results

presented in Fig. 10 shows close to linear scaling for up to 14976 PE in both nested and standalone runs. The difference in time per step between the nested and standalone runs can be interpreted as the additional computational time needed by the nesting algorithm. A jump in the time taken to compute one step is observed when more than 8192 PEs are used. This is a hardware dependent increase in communication time as the nodes are grouped as 'islands' on SuperMUC system at the Leibniz Super-

computing Centre. The communication within the nodes of the same island is faster than the communication across multiple islands. The strong scaling efficiency in Table 4 is calculated keeping the run with lowest number of PEs as the reference. As the number of grid points per PE is reduced from run A to E as shown in Table 5, the nested runs shows slightly better efficiency than the non-nested runs. The average time per step of the nested grid is 3 times higher than the non-nested set-up for run A but the factor decreases to about 2.5 for run E. This improvement is possibly due to reduction in waiting time between the FG

and CG as the number of grid points per PE decreases.

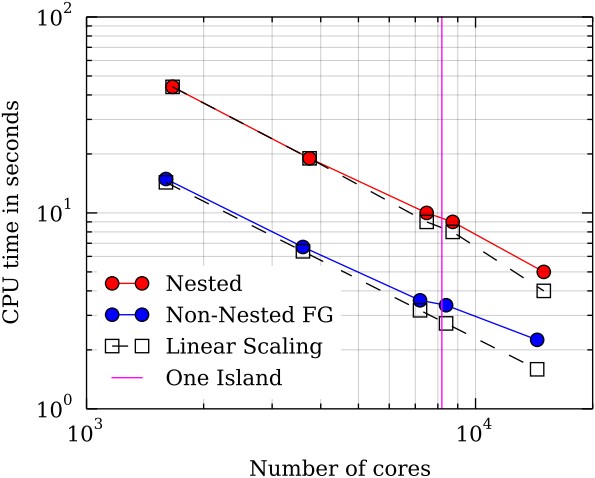

**Figure 10.** The nested simulations show close to linear scalability. A Non-Nested domain with same number of grid points as the FG is plotted to benchmark the scalability of the standard version of PALM on the same machine. The difference between the blue and the red line is approximately equal to the additional computational time needed by the nesting routines. The simulations were performed on SuperMUC at the Leibniz Supercomputing Centre. Each node has 32 GB of main memory and two Sandy-Bridge processors with 2.7 GHz, each processor has 8 cores (Anastopoulos et al., 2013).

### 3.4   Practical Considerations

In this paragraph we summarize some guidelines for using this nesting approach. In PALM, the user has the choice to se-lect between Wicker-Skamarock (Wicker and Skamarock, 2002) and Piacsek-Williams (Piacsek and Williams, 1970) for the advection scheme. Similarly, for solving the Poisson equation for the pressure, the user can choose between the FFT or Multi-

Grid based solver. During the development and the validation of the two-way nesting, only the Wicker-Skamarock advection scheme and FFT based pressure solvers were tested. The two-way nesting supports only periodic boundary conditions in the

horizontal for both CG and FG, and therefore an FFT based pressure solver is an appropriate choice. However, to be able to use Multi-Grid solvers, for e.g. in non-periodic horizontal boundary conditions, modifications to the two-way nesting algorithm will be needed. The large scale forcing feature in PALM is found to be compatible with the nesting algorithm without further modifications. Other features like canopy parameterization, radiation model, land surface models etc. have not been tested.

Our implementation of the vertical nesting allows only integer nesting ratios in all directions. The height of the nested domain has a direct influence on the accuracy of the two-way nesting algorithm. Based on our trials (not shown) we recommend that the FG covers at least 12 grid levels of the CG. For better computational performance we recommend that the number of grid points per PE in the CG is kept at only 40 to 80 percent of the FG value. The reduced work load of the CG is expected to minimize the waiting time of the FG during the concurrent time advancement by the quicker CG pressure solver step. However, the

actual improvement in performance will depend on the memory available, processor speed and the inter-node communication architecture of the computing cluster and the optimal load balancing can only be found through trials. Furthermore, the choice of the domain size is often restricted by the topology of the processor decomposition. In a 2D decomposition, the number of grid points along the x-direction should be an integer multiple of the number of PE along x and similarly for y-direction. This condition has to be individually satisfied for the CG and the FG.

Though our nesting technique makes resolving the surface layer resolution down to 0.5 m for a moderately large domain computationally feasible, care should be taken to ensure the validity of such LES. In PALM, the height of the first grid point should be at the least twice greater than the local surface-roughness parameter. This technical restriction is common to all models that employ MOST and ensures the proper evaluation of the logarithm needed in the calculation of $u^*$. Furthermore, Basu and Lacser (2017) recently recommended that MOST boundary-conditions should be adapted for very high-resolution

LES where the first grid point is smaller than 2-5 times the height of the roughness elements.

## 4   Summary

We presented a two-way grid nesting technique that enables high resolution LES of the surface layer. In our concurrently parallel algorithm, the two grids with different resolution overlap in the region close to the surface. The grids are coupled, i.e the interpolation of the boundary conditions and the feedback to the parent grid are performed, at every sub-step of the

Runge-Kutta time integration. The anterpolation of the TKE involves the Germano identity to ensure the conservation of total kinetic energy. The exchange of data between the two grids is achieved by MPI communication routines and the communication is optimized by derived datatypes. Results of the convective boundary layer simulation show that grid nesting improves the vertical profiles of variance and the fluxes in the surface layer. In particular, the profiles of the vertical temperature flux are improved. The current vertical nesting only works with periodic boundary conditions and with the same horizontal extent

in both the domains. The nested simulation needs 4 times less computational time than a full high resolution simulation for comparable accuracy in the surface layer. The scalability of the algorithm on up to 14976 CPUs is demonstrated.

## 5 Code availability

The PALM code is distributed under the GNU General Public License. The code (revision 2712) is available at https://palm.muk.uni-hannover.de/trac/browser/palm?rev=2712.

*Author contributions.* SH was the main developer of the model code, with FDR as side developer, SR supporting the code development and MM, SR and FDR supervising the development. The experiment was designed by SH, FDR, SR and MM and carried out by SH, who also performed the validation. Visualization was done by SH, and the original draft written by SH and FDR, with review and editing by SR and MM. Funding acquisition and administration by MM.

*Acknowledgements.* This work was conducted within the Helmholtz Young Investigators Group "Capturing all relevant scales of biosphere-atmosphere exchange – the enigmatic energy balance closure problem", which is funded by the Helmholtz-Association through the President's Initiative and Networking Fund, and by KIT. Computer resources for this project have been provided by the Leibniz Supercomputing Centre under grant: pr48la. We thank Gerald Steinfeld for sharing his original notes and code of a preliminary nesting method in PALM. We also thank Matthias Sühring and Farah Kanani-Sühring of the PALM group for their help in standardizing and porting the code, and we thank Michael Manhart for fruitful discussions.

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
