# Peer review of "Vertically nested LES for high-resolution simulation of the surface layer in PALM (version 5.0)"

_Geoscientific Model Development, 2018_

## Referee Comment (RC1) · Harris (Referee) · 4 Jan 2019

Review of Huq et al., "Vertically nested LES for high-resolution simulation of the surface layer in PALM (version 5.0)."

This is a very good paper on an oft-neglected topic and should be published. The issue of how to handle the vertical dimension on a nested grid is rarely addressed in the literature and this paper is a welcome addition that should help spur further discussion and development in this area. Many times in this article the authors either posed solutions to issues arising during grid refinement that I have myself encountered or gave excellent descriptions of technical details that relatively few papers on grid refinement address.

There are two particularly commendable components to the "anterpolation" (an excellent term!) from the fine to the coarse grid, both of which ensure that the nesting preserves appropriate physical quantities. The first is in Eq. 7, that the velocity components are averaged only over the faces shared between the nested and coarse grids and thereby ensuring that the volume-mean (non-)divergence is conserved by the anterpolation. (This idea is also used in Harris and Lin, 2013, MWR, although that paper uses a different grid staggering and thereby conserves a different cell-mean quantity.) The other is the "Germano Identity" of Eq. 8, which recognizes that resolved components on the nested grid may indeed be sub-grid on the coarse grid. I am unaware if other two-way nested LES models make this identification.

There are a few minor issues with this paper that I would like the authors to address before it is published.

The 5x refinement is quite aggressive and that the nesting artifacts are minimal shows how well the method works, at least for the problems considered here. The improvement to the profiles well above the grid interface is a particularly strong indication of how well the method works to improve the solution. However it is often hard to see artifacts in a snapshot such as Fig. 4. Would a time-averaged spatial plot show more artifacts? As seen in Fig. 6 there are some artifacts seen in the averaged profiles of the velocity variances, especially in the v variance. Why might v have larger shift in the variance between the two grids? Would these appear if a time-average of the variance were plotted as in Fig.4?

The issue of reflections of vertically-propagating gravity waves at the top boundary of a vertically-nested grid was considered by Clark and Farley (1984, JAS). In this model the nested upper boundary condition is relatively simple, which is OK for the Boussinesq LES problem presented here in which there are no sound waves and any vertically-propagating gravity waves would be very well-resolved. Do the authors expect that at coarser resolutions ($\sim$ 1 km) or if compressible equations are used that the form of the upper boundary condition used here would still yield acceptable results?

I found it strange that the two grids use the same timestep, which could introduce a significant computational burden. Furthermore the communication is done every single timestep, which also introduces substantial overhead due to the amount of message passing needed for the antepolation. Has any consideration been given to a longer communication timestep, or to use a longer timestep on the coarse grid?

Other comments:

- Sec 2.2: Quadratic interpolation is used for scalars. Does this introduce new extrema or negative values into the interpolated fields?

- Table 2: It seems strange that the SA-F run is more than 2000x more expensive than SA-C despite having only 125x more grid cells. Is this correct?

- Sec 3.1: The potential use of a sponge layer is briefly discussed. Do the authors plan to look more into this in future work to alleviate some of the artifacts at the upper boundary?

- The lines in Figures 5–7 are difficult to distinguish because they overlap so much. Perhaps thicker background lines for the SA simulations overplotted by thinner lines for the two grids of the nested grid would work better.

- Sec 3.4: The authors recommend an odd refinement ratio. Why would this be? The sort of averaging anterpolation used should be able to handle even refinements as well. Also it is said that the first five gridpoints in an LES are unreliable; why is this, and in which direction?

---

## Referee Comment (RC2) · Anonymous Referee #2 · 13 Jan 2019

This paper describes a methodology for utilizing a nested mesh with refined vertical grid spacing within the surface layer of the atmospheric boundary layer, to provide improved fidelity of boundary-layer flow simulation at a reduced computational cost, relative to utilizing fine vertical resolution throughout the entire domain. The paper carefully describes the numerical procedure for integrating the nested domain, including thorough discussions of boundary conditions (interpolation/anterpolation) and computational efficiency. The merits of the procedure are demonstrated by comparing snapshots of vertical velocity, and vertical profiles of various mean and turbulence quantities from simulations using coarse vertical mesh spacing throughout, simulations using fine vertical mesh spacing throughout, and finally simulations using fine mesh spacing only within the nested domain overlapping the surface layer, with coarser resolution above.

[Figure]

Nesting is seen to significantly improve the profiles within the nested region, while also more modestly improving some features farther aloft, with a substantial computational savings of approximately factor of 7. The algorithm also shows good scaling up to about 15000 cores.

Overall I find the paper to be very well written and informative, and think it will be make a valuable contribution to the literature. I have a few recommendations to address questions I had while reading that I think will further strengthen the paper, as itemized below.

Page 11, line 5. You normalized all the profiles using scaling quantity values from SA-F only, rather than values from the respective simulations. Are there any surprises or interesting features when scaling each profile with data obtained from their respective simulations?

A general comment for all of the vertical profile figures that is relevant here is to use different line styles, in addition to the different colors, to better differentiate profiles that are nearly on top of each other. With this strategy, you should be able to plot additional data without making the plots unwieldy to decipher.

Page 11, line 9: I think it would be interesting to see the anterpolated values, just to see how the algorithm is working behind the scenes. The same comment as above regarding plotting these additional data within the same plot applies here.

Page 12, lines 5-10 & Fig. 6. Please explain more thoroughly the discontinuities in all profiles between CG and FG near the FG top. Do the plotted profiles utilize the sponge layer that you describe, or not? Perhaps you could show the results with and without the sponge layer, using different linestyles and colors, as described above.

While you show mean profiles of various quantities, it would be nice to also see if there is any impact of nesting on the structures resolved within the CG above the FG in the nested simulations, relative to the SA-C (or within the nested domain relative

to SA-F, although this is not as relevant). Perhaps comparing spectra of streamwise velocity and/or w at a few heights would provide some useful information on this issue. If the nested FG in the surface layer is able to improve the instantaneous structures resolved within the CG above, that would be another noteworthy advantage of the vertical nesting capability.

―――――――――――――――――

---

## Referee Comment (RC3) · Anonymous Referee #3 · 23 Jan 2019

"Vertically nested LES for high-resolution simulation of the surface layer in PALM (version 5.0)"
by Huq *et al.*.

| | |
|---|---|
| Manuscript: | GMD-2018-287 |
| Title: | Vertically nested LES for high-resolution simulation of the surface layer in PALM (version 5.0) |
| Authors: | Sadiq Huq, Frederik De Roo, Siegfried Raasch, and Matthias Mauder |

**Recommendation**

Major revisions (but not very major)

Evaluation of Referee:

| | Excellent | Good | Fair | Poor |
|---|---|---|---|---|
| Scientific significance | x | | | |
| Scientific quality: | | x | | |
| Scientific reproducibility | | x | | |
| Presentation quality | | x | | |

**General**

- The paper describes a modification of the PALM large-eddy simulation code that enables vertical grid nesting in the layer adjacent to the Earth's surface. The resulting increase in resolution in the surface layer improves the representation of turbulence (since a smaller fraction of the turbulent motion has to be represented by the subgrid parameterization). With this technique, it becomes feasible to study turbulence in the surface layer while still resolving the full atmospheric boundary layer above it.

- The employed method builds on existing and established methods. The main contribution of the paper is that it describes the implementation of vertical nesting for a code that is publically available.

- The paper is well written and in general well structured.

However, I do have some comments:

a. I miss a thorough discussion on how the subgrid fluxes are handled at the interface between the course grid domain and the fine grid domain. I could imagine that the subgrid fluxes at the boundary between CG and FG would need to be interpolated. Or continuity of the subgrid fluxes at that interface could be ensured by the subgrid models on either side of the interface. However, I so not see how the subgrid flux between CG and FG are handled in a conserving way: what leaves the CG should enter the FG and the other way around.

b. The validation of the results of the nested simulation (characteristics of turbulent fields) is rather superficial (fluxed and variances, no spectral analysis or higher order moments; also little consideration for subgrid contributions).

c. The analysis of the timing of the simulations (scaling, overhead, net gain etc.) is limited.

d. The application of boundary conditions to the nested grid is insufficiently clearly described:

- Is the Dirichlet condition for horizontal wind components and scalars applied to a point

just above the fine grid domain, of to the highest point just inside the fine grid?

- The equations given for the interpolation algorithm lack explanation.

e. The structure of the introduction could be improved. After the overview of the history of LES, I would expect a clear definition of the problem (we need high resolution where it matters: close to the surface (and in the entrainment zone), an overview of how people have solved this until now, what is that we still not know/can/have?, and how are you going to solve it. Also the structure of section 2 could be improved to more clearly separate the different aspects of the new model.

Below I will provide detailed comments

*Note: in the comments below, the comment is preceded by the page number-line number.*

**Detailed comments**

1. 2-27: You immediately make the jump to grid-nesting. However, the main point is that you need increased resolution. And if you cannot afford to increase the resolution in the entire domain, you want to do it locally. *One* way of doing that is by grid-nesting. But there are other ways: if one does not insist to stick to a structured grid, local grid refinement (without nesting) is feasible. This grid refinement can even be made dependent on the flow itself (see van Hooft *et al.*, 2018). So: grid-nesting is just one of the ways to locally increase resolution.

2. 3-7 to 16: here you explain why vertical nesting is needed. But you started that argument already in line 2-28 to 30. Please restructure your argumentation (either move 3-7 to 16 to the point where you introduce vertical nesting (and then talk about horizontal nesting to show what we know from that), or first introduce horizontal nesting and then make the step to vertical nesting (or ignore horizontal nesting altogether, since vertical nesting is in itself not new, just your implementation in PALM is new).

3. 3-17: it seems that Clark and Hall (1991) deals with horizontal nesting. To what extant is it still relevant for this paper?

4. 3-23: '...superior when the waves ...': doe you mean 'when' or 'if'? And what happens if/when the waves are not well resolved? In what way is this relevant for the present paper on the simulation of turbulence?

5. 3-28: '.... both the resolved and SGS fluxes...': does this also hold for the finite difference code used here? In what way would/does it increase coding complexity?

6. 4-17: please explain the variables used in the equations. In particular the notation for resolved variables and subgrid variables is important. Furthermore, I assume you include the tendency equation for potential temperature because the potential temperature plays a role in the SGS-TKE equation and in the momentum equation. But then you should also include the moisture tendency in order to be able to determine the tendency of the virtual potential temperature (which then also should be used in the buoyancy terms). Finally, the heat flux that appears in equation (4) is the subgrid heat flux: (1) apparently you denote subgrid variations by a single prime and the filtering operation by an overbar and (2) in the model the subgrid heat flux is parameterized using a gradient hypothesis (also the next term, the transport term, is parameterized).

7. 4-20: 'guarantees a stable': how does the choice of the time integration method guarantee a stable solution. The magnitude of the time step would still play a role (and it does, as later on you invoke the CFL criterion). So why mention stability here?

8. 4-23: I assume that you refer the *vertical* zero pressure gradient here.

9. 5-9: apart from updating the ghostpoint, there is also global communication needed in the

Poisson solver. This involves way more communication than the ghostpoint update.

10. 5-10: regarding the structure of the rest of section 2: I would suggest to restructure this section as follows:

    2.2 Model structure

    2.2.1 Grid configuration (now 2.2, up to line 5-29)

    2.2.2 Nesting algorithm

    2.3 Translation between grids (line 5-9 until7-4)

    2.3.1 Anterpolation

    2.3.2 Interpolation

    2.4 Parallel inter grid communication

    (after 2.2.2 it is clear where and why anterpolation and interpolation are needed).

11. 5-30: only the vertical velocity really has a boundary *at* the top of the FG. For the other velocity components and scalars it is unclear whether the boundary condition (interpolation from CG) is applied to a ghost point (just above the FG) or to the first point just below the boundary of the FG.

12. 5-32: what is the 'logical' direction? If figure 1 would be upgraded (see below), this 'logical' linear interpolation would probably become clear.

13. Figure 1: the current figure is not very informative. I would suggest to replace it by a figure in which you show a few CG cells as well as the FG cells within one or two of them (preferably with a grid ratio of 3, not more). Then clearly show how the interpolation of vertical velocities, as well as horizontal velocities and scalars works (in order to support the interpretation of equations (5) as well as the notion that the velocities are interpolated in a 'logical' direction. The connection to equations (5) could also clarify the meaning of the various indices (lowercase and uppercase).

14. Please completely rework the equations and add explanations:

    - Make clear that the first equation is the actual interpolation, and all the other equations just define the various parameters occurring therein.

    - In which coordinate direction does $i$ vary: only in the $x$-direction, or also in other directions. Or are we actually looking at a 2D or 3D stencil of which only one dimension is shown?

    - The capital indices $I$, $J$, and $K$ are counting through the entire domain, I assume. But how about the lower case indices: do they start counting at 1 (or zero) within each CG cell, or do they also count globally?

    - In the $2^{nd}$-$4^{th}$ equations you introduce $H_k$. What is the value of the index $k$. Or does the repeated index imply summation? If so, what is the range of values that $k$ can take: 1, 2 and 3 because of the dimensionality, or 1, 2, ... $n_x$ because of the number of FG cells in a CG cell?

15. 6 - equation (6): what is the range of values for $i, j$ and $k$ ? Is there a mapping that gives the global $i,j,k$ values for a given $I,J,K$ or, are these local $i,j,k$ values, running as 1, 2, ... $n_x$ ?

16. 7 - equation (7): idem

17. 8-7 and 11: please keep the discussion on the solution of the Poisson equation in one place. What is the value of the pressure gradient that is imposed as a Neumann boundary condition? If it is zero, reflections could occur, but if you use something non-zero: how do you determine the value of this gradient? Is it derived from the CG pressure field?

18. 8-12 and 13: please clarify how the value of the imposed pressure gradient is determined/chosen.

19. 8-31: '... the higher number of PE available in the FG.': this is stated as if the reader already knows that there are more PEs in the FG (although for any grid ratio above 2 it is indeed logical that the number of FG PEs is larger than the number of CG PEs). But in addition, it

      is unclear to me why the higher number of FG PEs would be relevant for the FG-to-CG communication.

20. 9-2 'should be kept lower': please explain the logic of this statement. I assume that the idea is that you want to reduce the total amount of idle CPU time on the FG PEs (*N* cores x wait time), which can be achieved by under-utilization of the (only) *M* cores running CG (better waste time on a few CG cores than on many FG cores). In order to know how this plays out in practice, you should show in your results the amount of time spent in the various steps in a RK substep: which fraction (and absolute time) of a time step is devoted to which substep in figure 2, and how much of this time is wasted time.

21. 9-12 'Dirichlet condition': to which values are the velocities set: zero for vertical wind and geostrophic for horizontal?

22. 9-13: what is the imposed temperature gradient at the surface?

23. 9-15 and 16: is the wind profile interpolated *linearly* from zero at the surface to geostrophic at the top? Does this out-of-balance initialization lead to an inertial oscillation?

24. Table 1: what is the boundary condition for wind? MOST with an imposed roughness length (what is the value) or an imposed stress?

25. Table 2:
   - please add the number of time steps needed to complete this simulation (in that way the reader can easily determine the time spent per gridpoint per time step.
   - For the reader it would also helpful to include the number of grid points per PE and the CPU time per grid point (but both numbers *can* be derived from the available data, so the reader could do it for themselves).
   - the number of cores devoted to CG and FG respectively is not motivated. Whereas later on you advise to assign to a CG PE 40-80% of the number of grid points that is assigned to a FG PE, here you use a fraction of 16%.
   - it is unclear to what extent the PE's are saturated in terms of memory usage: could this problem be run on even a smaller number of processors to improve performance?
   - Please include information on the time (absolute and/or as a fraction) that is used waiting for input from CG to FG or the other way around. This would be helpful to determine the optimal division of labor between CG PEs and FG PEs (in terms of grid points per node).

26. 10-2: what initial perturbation is applied to get turbulence started? How did you verify that after 9000 s the flow was in equilibrium?

27. 11-5: part of the ingredients for the scaling variables are in fact imposed boundary conditions (the surface heat flux), whereas indeed another part (the surface shear stress) results from the flow (and hence need to be derived from one of the simulation results (assuming that a roughness length is prescribed).

28. 11-6 and 8: surface heat flux in the expression for $w*$: overbar is missing and this is not a turbulent flux (so do not use a covariance flux).

29. 11-8: although it will not change the lines in the graph, normalizing the temperature with the surface value is very illogical. Please plot the temperature with some reference value (e.g. the surface value) subtracted and normalized with $\theta*$.

30. 12-1: what would/could be the mechanism that makes that the higher resolution in the surface layer would affect the variance profile well above the FG domain?

31. 12-3: please use the same scaling variable for all velocity components! If not, the different variances (which together constitute the turbulent kinetic energy) cannot be compared. Furthermore, the given flow is close to free convection, so using the friction velocity as a scaling variable does not make sense.

32. 12-5: you refer to an overshoot in the *v* variance. The *u* variance shows an overshoot as well.

I assume that the profiles shown are based on the resolved variances only. In that case, we should keep in mind that in the CG domain a larger proportion of the TKE is contained in the subgrid scales. Could this explain the jump? Please include an analysis of the difference in SGS-TKE between the two domains at the top of the FG (of course there is the difficulty of separating the SGS TKE into the three components, but at least quasi-quantitatively such an analysis could shed light on these jumps/overshoots.

33. 12-7: how would the anterpolation influence the vertical velocity variance in the FG domain. Please explain the/a mechanism. Or is it a result of the fact that the upper boundary conditions for pressure at the top of the FG is not well-defined?

34. 13 figure 6: the variance profiles give some information on the quality/realism of the simulated turbulence. One analysis that is missing (related to the point made above regarding the overshoot) is whether the increased resolved TKE is the amount that would be expected based on the increased resolution (and hence reduced reliance on the subgrid model). To properly analyse that one would need turbulent spectra to see how much kinetic energy is contained in the additionally resolved scales.
Additionally, spectral analysis (preferably with 2D spectra) would help to show to what extent the extra resolved turbulence has the expected turbulent characteristics (increased variance is nice, but does not need to be additional turbulence, it could also be increased noise).

35. 13-1: The heat flux profile is not the prime quantity at all! For a quasi-stationary convective boundary layer with imposed surface flux the heat flux profile is the most boring part of the simulation. Provided that the entrainment flux is represented well, the flux profile is by definition linear, varying between the imposed surface flux (so no surprises there) and the entrainment flux (which, admittedly, needs to be represented correctly by the simulation: still some freedom there). This linear flux profile is completely independent of the quality and resolution of the simulation. The only freedom there is is which part of that flux is carried by the resolved scales and which part is carried by the subgrid model. Hence the perfect correspondence between all simulations (full FG, full CG, nested CG and nested FG). Hence, please do not use the heat flux profile as a measure of the quality of the simulation.

36. 14-4: '...we increase the resolution further': do you mean to increase the grid ratio, the size of the FG region, or the overall resolution of the CG domain?

37. 14-9: 'in terms of communication time': do you only look at communication time because that is the most restricting, or because you are only interested in that (in this context)? And why should the number of domains be equal in x and y direction: please explain the logic of this (and does it also hold if the length of the domain is different in x and y direction?

38. 15-1 to 5: why is the setup of these simulations (in terms of the total number of points and ratio of number of grid points between CG cells and FG cells) so different from the original runs? Are the performance results still relevant to understand those first runs? If so, why? Please give the setup of these runs in a table similar to table 2 (not 'number of grid points is *around...*').

39. 15- Figure 8:
- on a log-log scale everything looks nice. Please give a more informative representation. E.g. use the strong scaling efficiency, which will vary between 1 and somewhere below 1 (for your data, using the left-most simulation as a reference, the efficiency goes down to about 90% for the right-most. But the question is, what would have been the CPU time for the smallest possible number of processors on which this case could have been run (memory-wise).
- In addition, find a more informative way to quantify the waiting time overhead.

"Vertically nested LES for high-resolution simulation of the surface layer in PALM (version 5.0)" by Huq *et al.*.

40. 16-9: 'large scale forcing .... compatible': do you refer to the large scale forcing in terms of pressure gradient/geostrophic wind? Or another large scale forcing? Why would it, or would it not, be compatible. Please clarify.
41. 16-12: 'accuracy': for accuracy in what sense (interpolation errors, truncation errors, turbulence statistics, stability, ....) should the grid ratio be not too large?
42. 16-14: 'first five grid points are unreliable': for which variables does this hold, in which aspect are the grid points unreliable (I assume you mean '*the results* at the first five grid points vertically displaced from the surface'): turbulence characteristics, mean profiles, noise, ....? Do you have a reference for this bold statement?
43. 16-16: I would like to see a quantitative motivation for this 40-80%.

**Very detailed comments**

1. 2-14: 'possible, by the time' → 'possible. By the time'
2. 2-18: 'supercomputers': also the people before Kröniger et al. used supercomputers. So remove 'with the help of supercomputers'.
3. 2-19: remove 'speeds'
4. 2-21: 'higher detail': 'higher' than what/when/who?
5. 2-28: 'Nesting has been applied...': because the previous sentence talks about vertical nesting, the reader may think that this sentence gives examples of that. But then at the end it turns out to talk about *horizontal* nesting. Please rephrase.
6. 3-3/4: 'techniques *are* ..... but often *uses* ...': 'uses' should be 'use' (plural)
7. 3-18: '... CG, there ...' → '... CG, and there ...'
8. 3-24: make explicit that the 'two different approaches' only refer to the 'anterpolation' mentioned in the sentence before. Furthermore, nothing is said –explicitly- about the 'pressure deficit correction' ("there are two types of cars: blue cars")
9. 4-9: 'additional equation': additional to what? The SGS-TKE equation *is* in the Deardorff method, so it is *not* additional to his work.
10. 4-11: 'The prognostic ...': move this sentence to below the equations (only after having presented the equations you need to talk about their discretisation).
11. 5-17: 'the grids' → 'grids' (this occurs in multiple places, please check.
12. 5-18: please explain here already that uppercase symbols refer to CG and lowercase symbols to FG.
13. 6-1: 'similar interpolation': in which way is it similar, and which way is it different?
14. 6-5: 'scalars' → 'CG scalars'
15. 6-5: 'The scalars .... corresponding FG scalars (eq. 6)'. How much more are you saying than 'An average is an average'. If you want to state more, please make that clear and explicit.
16. 7-6: 'We implement....'. Well, that does not really come as a surprise: you gave that away already (see my suggestions for an alternative structure for section 2).
17. 8-7: '... is also updated...'. What else is updated? You mean the pressure? And is the vertical velocity updated throughout the FG, or are you only referring to the vertical velocity at the CG-FG interface?
18. 8-20: '...process. Whereas....' → '...process, whereas....'
19. 8-20: 'exchange' → 'exchange*d*'
20. 8-23: 'local PE's 2D processor co-ordinate': in what way is the PE different from the processor? → 'local 2D processor (or PE) coordinate'
21. 9-11: 'is set to' → 'has'
22. 10, Tables 1 and 2: please format the tables properly as tables should be formatted (including column headings and a consistent demarcation of rows and columns)

"Vertically nested LES for high-resolution simulation of the surface layer in PALM (version 5.0)" by Huq *et al.*.

Table 1: please note that the surface heat flux is not a turbulent flux (there is no vertical velocity (variation) *at* the surface. Furthermore, even if you would like to denote it as a turbulent flux, please add an overbar.

23. 11, figure 4: the lower panel is −vertically- not exactly to scale with the area indicated with the dashed line in the upper panel.
24. 11-2: 'flux profiles' → 'fluxes'
25. 11-3: the given expressions are not fluxes, but products of resolved deviations: please include an averaging operator to make it a flux.
26. 11-13: 'at the boundary layer height' → 'at the top of the boundary layer'
27. 12-1: 'An one-way' → 'A one-way'
28. 12-5: 'variance seen' → 'variance can be seen'.
29. 14-5: 'Simulations with O(1) ...': are you referring to that resolution for a full domain, or only for the FG part of a nested simulation? In fact, it is unclear where you are heading with lines 14-2 to 14-6.
30. 14-13: 'new nested simulation': new relative to? I assume that you mean new relative to the runs described in tables 1 and 2. These new simulations were made for the performance test only?
31. 16-4: 'Poisson equation' → 'the Poisson equation'
32. 16-7: 'FFT' → 'an FFT'
33. 16-28: 'energy conserving methods': I have not seen that term earlier in the paper. Where was this discussed before? Or are you referring to the anterpolation of SGS TKE? In that case, please be a bit more explicit.
34. '... optimized for performance': how were they optimized, where can I read about that optimziation?

**References**

van Hooft, J.A., Popinet, S., van Heerwaarden, C.C. et al. (2018). Towards Adaptive Grids for Atmospheric Boundary-Layer Simulations. *Boundary-Layer Meteorol* **167**: 421-443. https://doi.org/10.1007/s10546-018-0335-9

---

## Author Comment (AC1) · 16 Apr 2019

**Response to Referee Comment 1 (Dr. Lucas Harris)**

It is often hard to see artifacts in a snapshot such as Fig. 4. Would a time-averaged spatial plot show more artifacts?

We agree that the artifacts will not be clearly visible in a snapshot. However, we included the snapshot to qualitatively show that the flow structures can propagate across the grid interface without any distortion and also show that smaller structures are better resolved in the FG. In the figure below, we show a vertical velocity cross-section time-averaged over 1800 s at the end of the 10800 s simulation. There are no visible artifacts.

[Figure]

As seen in Fig. 6 there are some artifacts seen in the averaged profiles of the velocity variances, especially in the v variance. Why might v have larger shift in the
variance between the two grids? Would these appear if a time-average of the variance
were plotted as in Fig.4?

We realized that in the initial runs we had failed to include the compiler flag (-fpmodel strict) necessary to make every realization deterministic. As a result, the plots in our initial submission have compiler optimization related random effects, that were noticeable in the variance profile. The results of both the nested and the standalone simulation were affected. We have now ensured that the simulations are deterministic, which improves their comparability. The plots and their description have been updated accordingly.

In the updated spatial velocity variance profile, we observe a marginal shift in u. For, clarity, the profiles

of the variance in the manuscript are based on the spatial variance. We cannot use the same spatial variance for visualizing the variability over an XY cross-section of the variance because the local information is already averaged out to obtain the spatial variance. However, we used a custom user code to output the 30 min averaged temporal variance of u shown below, and we can plot the spatial variability of this quantity. The shifts that are observed in the variance profiles are not noticeable in the contour plot. Since the custom user code has an influence on the computational performance an additional run with identical set-up described in the manuscript was performed to produce this output.

[Figure]

Temporal variance of u averaged over 30 mins.

The issue of reflections of vertically-propagating gravity waves at the top boundary of a vertically-nested grid was considered by Clark and Farley (1984, JAS). In this model the nested upper boundary condition is relatively simple, which is OK for the Boussinesq LES problem presented here in which there are no sound waves and any vertically-propagating gravity waves would be very well-resolved. Do the authors expect that at coarser resolutions (~ 1 km) or if compressible equations are used that the form of the upper boundary condition used here would still yield acceptable results?

In our implementation of the nesting method, we assume that most of the TKE is resolved well down to the inertial subrange, except for the lowest few grid layers. This allows us to use zero-gradient Neumann boundary condition for TKE at top of the nested grid. This assumption will not be valid at coarser resolutions (~1 km) and therefore such simulations will not be possible with our method. Furthermore, an advanced sponge layer should also be implemented. We have updated the TKE boundary condition in the manuscript:

"Furthermore, in our implementation of the nesting method, we assume that most of the TKE is

resolved well down to the inertial subrange, except for the lowest few grid layers. This allows us to use zero-gradient Neumann boundary condition for TKE at the FG top boundary."

I found it strange that the two grids use the same timestep, which could introduce a significant computational burden. Furthermore the communication is done every single timestep, which also introduces substantial overhead due to the amount of message passing needed for the antepolation. Has any consideration been given to a longer communication timestep, or to use a longer timestep on the coarse grid?

Initially we had considered allowing a longer timestep on the coarse grid. However, we did not implement the idea due to parallelization restrictions. We have used a simple parallelization approach where the CG and FG are each assigned a dedicated group of processors. Allowing for a longer timestep on coarse grid also leads to an increased CPU idle time on the CG PE. Therefore, we restricted both the grids to have same timestep. Communicating at every single timestep indeed increases the computational cost. We have not considered limiting the anterpolation to every few time integration steps.

Other comments:

- Sec 2.2: Quadratic interpolation is used for scalars. Does this introduce new extrema or negative values into the interpolated fields?

We agree that quadratic interpolation could introduce new local extrema. In our tests we found that the extrema of the interpolated FG scalars had less than 0.005 % difference compared to the CG scalars at the nesting height. However, the horizontal mean of the CG and the FG is equal. We chose the quadratic interpolation of Clark and Farley (1984) because in the interpolation formulation, the coefficient alpha is chosen such the conservation condition of Kurihara et al. (1979) is satisfied and consequently the interpolation is reversible. We have updated the manuscript as:

"The same interpolation formulation is also used to initialize all vertical levels of the fine grid domain at the beginning of the nested simulation. The interpolation is reversible as it satisfies the conservation condition of Kurihara et al. (1979):"

$$< \phi > = < \Phi > \tag{9}$$

- Table 2: It seems strange that the SA-F run is more than 2000x more expensive than SA-C despite having only 125x more grid cells. Is this correct?

The SA-F has more grid cells and consequently needs more cores, the number of grid points per core in SA-F is more than 6 times higher than SA-C. Furthermore, the higher resolution in SA-F demands smaller time steps than SA-C. The total number of time steps in SA-F is 5 times more than SA-C. Therefore, the SA-F run is indeed 1000 times more expensive.

We have updated the grid configuration Table 3 to include the grid points per core and the time steps needed in each simulation.

- Sec 3.1: The potential use of a sponge layer is briefly discussed. Do the authors plan to look more into this in future work to alleviate some of the artifacts at the upper

boundary?

Our simplified approach provides reasonable results. Nevertheless, we agree that a thorough investigation of a sponge layer certainly needs more attention to effectively alleviate the artifacts at the upper boundary. We have not planned a thorough investigation of the sponge layer in the near future. However, the main developers of PALM are developing 3D nesting in PALM (similar to our method) and we expect that  the artifacts at the boundary will be investigated further.

- The lines in Figures 5–7 are difficult to distinguish because they overlap so much.
Perhaps thicker background lines for the SA simulations overplotted by thinner lines for
the two grids of the nested grid would work better.

The Figures 5–9 have been updated. As suggested, thicker dashed lines for the standalone simulations and thinner lines for the two nested grids improve the visibility of the overlapping lines.

- Sec 3.4: The authors recommend an odd refinement ratio. Why would this be? The
sort of averaging anterpolation used should be able to handle even refinements as well.

The averaging anterpolation can indeed handle both odd and even refinements. However, if the grid nesting ratio is odd, there will be one FG cell of which the center is exactly at the same position as the center of the coarse cell as shown in the updated Figure 1 in the manuscript, due to the Arakawa-C grid in PALM. Such a nested grid set-up is expected to increase the numerical accuracy of the anterpolation operation.

Also it is said that the first five gridpoints in an LES are unreliable; why is this, and in
which direction?

We meant the first five grid layers in the vertical direction form the surface are not reliable because a lot of turbulence is still sub-grid. This is only based on our experience, due to lack of literature to support this point we have removed this sentence.

**Response to Referee Comment 2**

Page 11, line 5. You normalized all the profiles using scaling quantity values from SA-F only, rather than values from the respective simulations. Are there any surprises or interesting features when scaling each profile with data obtained from their respective simulations?

[Figure]

The variances of velocity and temperature normalized by the values from the respective simulations are shown below. Since the boundary layer height differs only about 30 m, the difference in the surface potential temperature is less than 1 K and the surface heat flux is constant between the simulations there is no considerable difference in the profiles normalized by SA-F values and the profiles normalized by the values from the respective simulations.

A general comment for all of the vertical profile figures that is relevant here is to use different line styles, in addition to the different colors, to better differentiate profiles that are nearly on top of each other. With this strategy, you should be able to plot additional data without making the plots unwieldy to decipher.

The Figures 5–9 have been updated. As suggested, different lines styles are now used, marginally thicker dashed lines for the standalone simulations and thinner lines for the two nested grids improve the visibility of the overlapping lines.

Page 11, line 9: I think it would be interesting to see the anterpolated values, just to see how the algorithm is working behind the scenes. The same comment as above

regarding plotting these additional data within the same plot applies here.

The variances of velocity and temperature close the surface are shown below. Compared to SA-C and SA-F, the CG profile in the anterpolated region is improved close to the surface.

We have added a profile of the sub-grid scale TKE to the manuscript, where the effect of maintaining the Germano identity during anterpolation is clearly noticeable. The anterpolated values are also shown in the vertical velocity skewness plot.

[Figure]

Page 12, lines 5-10 & Fig. 6. **Please explain more thoroughly the discontinuities in all profiles between CG and FG near the FG top.** Do the plotted profiles utilize the sponge layer that you describe, or not? Perhaps you could show the results with and without the sponge layer, using different linestyles and colors, as described above.

The plotted profiles use our simplified sponge layer approach, i.e. limiting the anterpolation to one CG cell less than the nested height. In our implementation, it is currently not possible to disable the simplified sponge layer as the data exchange from FG to CG for anterpolation are by default limited to one CG cell less than the nested height. We have expanded the explanation as:

"The v and w FG profiles have a better agreement with the SA-F than the u variance. The u and v variance in Fig. 7 (d and e) lie between SA-C and SA-F indicating that the resolved variances are improved compared to the SA-C but not sufficiently resolved to match SA-F. At the nesting height the variances deviate more from the SA-F and approach the CG values. Due to conservation of total kinetic energy across the nest boundary more CG TKE is contained in the sub-grid scale. Consequently, the resolved CG variances could have an undershoot as compared to SA-F, resulting in an undershoot of the FG variances too at the nesting height. Above the nesting height, the variance of u, v and w in CG are similar to SA-C."

While you show mean profiles of various quantities, it would be nice to also see if there is any impact of nesting on the structures resolved within the CG above the FG in the nested simulations, relative to the SA-C (or within the nested domain relative to SA-F, although this is not as relevant). **Perhaps comparing spectra of streamwise velocity and/or w at a few heights would provide some useful information on this issue.** If the nested FG in the surface layer is able to improve the instantaneous structures resolved within the CG above, that would be another noteworthy advantage of the vertical nesting capability.

We have added new figure to compare the spectra of SGS TKE and vertical velocity at two heights, one within the nested grid and one above the nested grid. The w spectra of CG above the nested height follows SA-C and improvement due to higher resolution at the surface layer is not noticeable. We added the following text to the manuscript:

"The horizontal spectra of SGS turbulent kinetic energy and vertical velocity are plotted in Fig. 9 at two levels, one within the nested grid and one above the nested grid height. The FG TKE spectra in Fig. 9 (c) perfectly overlaps the SA-F spectra. The CG spectra has higher energy than the SA-C, this corresponds to the higher CG TKE values observed in Fig. 8 (c). As the limit of the grid resolution is reached at high wavenumber, the drop in the CG spectra is marginally delayed compared to SA-C. This improvement at high wavenumber is due to feedback from the FG. Similarly, the vertical velocity spectra in Fig. 8 (d) shows marginal improvement at high wavenumber for the CG. While the FG agrees with SA-C at high-wave number and at the spectra peak, at low wavenumber FG follows the CG spectra. At the level above the nested grid, the CG spectra agrees with SA-C for both TKE and the vertical velocity."

**Response to Referee Comment 3**

**I miss a thorough discussion on how the subgrid fluxes are handled at the interface between the course grid domain and the fine grid domain. I could imagine that the subgrid fluxes at the boundary between CG and FG would need to be interpolated.**

The PALM model employs a 1.5 order turbulence closure parameterization. Therefore, at each time step, all the sub-grid fluxes are derived from the turbulent kinetic energy and the resolved gradients. Therefore, it suffices that the prognostic variables are communicated correctly. In our implementation of the nesting method, we assume that most of the TKE is resolved well down to the inertial subrange, except for the few lowest grid-layers. This allows us to use zero-gradient Neumann boundary condition for TKE at top of the nested grid.

We have added the following sentences to the manuscript:
"In the 1.5 order turbulence closure parameterization, all the sub-grid fluxes are derived from the turbulent kinetic energy and the resolved gradients at each time step. Therefore, the sub-grid fluxes do not have to be interpolated from CG to FG at the top boundary. Furthermore, in our implementation of the nesting method, we assume that most of the TKE is resolved well down to the inertial subrange, except for the lowest few grid layers. This allows us to use zero-gradient Neumann boundary condition for TKE at the FG top boundary."

Or continuity of the
**subgrid fluxes at that interface could be ensured by the subgrid models on either side of the interface**. However, I so not see how the subgrid flux between CG and FG are handled in a conserving way: what leaves the CG should enter the FG and the other way around.

The sub-grid fluxes do not need to be continuous, only the sum of resolved and sub-grid fluxes should be. In the FG, there is a larger proportion of resolved flux (due to the smaller grid spacing) and less sub-grid flux.

However, we agree with the reviewer that for LES that employ higher order closure models (where the fluxes are independent variables) the sub-grid fluxes should be communicated as well.

b. The validation of the results of the nested simulation (characteristics of turbulent fields) is rather superficial (fluxed and variances, no spectral analysis or higher order moments; also little consideration for subgrid contributions).
We have included spectral analysis and vertical profiles of vertical velocity skewness and SGS TKE to expand our analysis. More information is provided in the detailed comments below.

c. The analysis of the timing of the simulations (scaling, overhead, net gain etc.) is limited.

We demonstrate the linear scalability of the nested simulations on more than fourteen thousand CPUs. However, in our benchmark runs, we did not profile the time taken by each nesting routine, and therefore a detailed analysis of the overhead is not possible. To overcome this limitation in our analysis, we executed standalone simulations with the same number of grid points as in the FG domain. The difference between the nested and the standalone in the scalability plot provides a rough estimate of the overhead. The analysis is expanded with the strong scaling efficiency suggested in the detailed comments.

d. The application of boundary conditions to the nested grid is insufficiently clearly described:

- Is the Dirichlet condition for horizontal wind components and scalars applied to a point just above the fine grid domain, of to the highest point just inside the fine grid?

We define 'top of the FG' as the highest point in the FG. This is the boundary point which is excluded from the CFD calculations. We added the following sentence in section 2.3.1:

"We define the top of the FG as the boundary level just above the prognostic level of each quantity."

- The equations given for the interpolation algorithm lack explanation.

The explanations to interpolation equations is added to the text. More information is provided in the detailed comments below.

e. The structure of the introduction could be improved. After the overview of the history of LES, I would expect a clear definition of the problem (we need high resolution where it matters: close to the surface (and in the entrainment zone), an overview of how people have solved this until now, what is that we still not know/can/have?, and how are you going to solve it. Also the structure of section 2 could be improved to more clearly separate the different aspects of the new model.

We have improved the structure of the introduction and re-structured section 2 as suggested. The changes are listed in the relevant detailed comments.

**Detailed comments**

1. 2-27: You immediately make the jump to grid-nesting. However, the main point is that you need increased resolution. And if you cannot afford to increase the resolution in the entire domain, you want to do it locally. One way of doing that is by grid-nesting. But there are other ways: if one does not insist to stick to a structured grid, local grid refinement (without nesting) is feasible. This grid refinement can even be made dependent on the flow itself (see van Hooft et al., 2018). So: grid-nesting is just one of the ways to locally increase resolution.

We have re-structured the paragraph to clearly define the problem and then introduce solutions other than nesting and then introduce a summary of the nesting literature. We updated the manuscript as:

"Still, especially in heterogeneous terrain, near topographic elements, buildings or close to the surface, the required higher resolution is not always attainable due to computational constraints. In spite of the radical increase in the available computing power over the last decade, large-eddy simulations of high Reynolds number atmospheric flows with very high-resolution in the surface-layer remain a challenge. Considering the size of the domain required to reproduce boundary-layer scale structures, it is computationally demanding to generate a single fixed grid that could resolve all relevant scales satisfactorily. Alternatively, local grid refinement is possible in the Finite-Volume codes that are not restricted to structured grids. Flores et al. (2013) developed a solver for the OpenFOAM modelling framework to simulate atmospheric flows over complex geometries using an unstructured mesh approach. Van Hooft et al. (2018) demonstrated the potential of adaptive mesh refinement technique where the tree-based Cartesian grid is refined or coarsened dynamically, based on the flow structures."

2. 3-7 to 16: here you explain why vertical nesting is needed. But you started that argument

already in line 2-28 to 30. Please restructure your argumentation (either move 3-7 to 16 to the point where you introduce vertical nesting (and then talk about horizontal nesting to show what we know from that), or first introduce horizontal nesting and then make the step to vertical nesting (or ignore horizontal nesting altogether, since vertical nesting is in itself not new, just your implementation in PALM is new).

We restructured the text to first introduce horizontal nesting and then focus our discussion on the vertical nesting. We would like to retain the discussion on horizontal nesting as our vertical nesting is motivated by literature in horizontal nesting. The vertical nesting discussion is restructured as:

"For our purposes, we will focus on vertical nesting, i.e. we consider a Fine Grid nested domain (FG) near the lower boundary of the domain, and a Coarse Grid parent domain (CG) in the entire of the boundary layer. While the latter's resolution is sufficient to study processes in the outer region where the dominant eddies are large and inertial effects dominate, such coarse resolution is not sufficient where fine-scale turbulence in the surface layer region is concerned.
The higher resolution achieved by the vertical nesting will then allow a more accurate representation of the turbulence in the surface layer region, by resolving its dominant eddies. For studies that require very high resolution near the surface (e.g. virtual tower measurements, wakes behind obstacles, dispersion within street canyons for large cities) a nesting approach is an attractive solution due to the reduced memory requirement. The challenge of a vertically nested grid is that the FG upper boundary conditions need to be correctly prescribed by the CG. Though vertical nesting is less common than horizontal nesting, it has been implemented in some LES models. A non-parallelized vertical nesting was explored by Sullivan et al. (1996) but this code is not in public domain and we could not find any record of further development or application of this code in publications. An LES-within-LES vertical nesting is implemented by Zhou et al. (2018) in the Advanced Regional Prediction System (ARPS) model."

3. 3-17: it seems that Clark and Hall (1991) deals with horizontal nesting. To what extent is it still relevant for this paper?

Though Clark and Hall (1991) deals with horizontal nesting, their error analysis of the nesting procedures is relevant in understanding the 'post-insertion' and 'pressure defect correction'. Their work has also provided motivation for other vertical nesting development (Sullivan et al. (1996). Updated the manuscript as:
"Clark and Hall (1991) studied two different approaches for updating the CG values, namely "post-insertion" and "pressure defect correction".
The two approaches were also investigated by Sullivan et al. (1996) in their vertical nesting implementation.

4. 3-23: '...superior when the waves ...': doe you mean 'when' or 'if'? And what happens if/when the waves are not well resolved? In what way is this relevant for the present paper on the simulation of turbulence?

Harris and Durran (2010) observed that only for moderately well resolved waves, the two-way interaction performed better than the one way interaction. Modified the text to introduce the concept of 'sponge boundary condition':

"Harris and Durran (2010) used a linear 1D shallow-water equation to study the influence of the nesting method on the solution and found the two-way interaction to be superior if the waves are well resolved. They introduce a filtered sponge boundary condition to reduce the amplitude of the reflected

wave at the nested grid boundary.”

5. 3-28: ‘.... both the resolved and SGS fluxes...’: does this also hold for the finite difference code used here? In what way would/does it increase coding complexity?

We had written:

“Sullivan et al. (1996) report that in the case of their Pseudo-Spectral LES, both the resolved and SGS fluxes need to be anterpolated to the CG and such a procedure increases coding complexity.”

Since no explanation for the increase in the coding complexity is found in the literature and also because this does not hold for the finite difference code, we have removed this statement.

6. 4-17: please explain the variables used in the equations. In particular the notation for resolved variables and subgrid variables is important. Furthermore, I assume you include the tendency equation for potential temperature because the potential temperature plays a role in the SGS-TKE equation and in the momentum equation. But then you should also include the moisture tendency in order to be able to determine the tendency of the virtual potential temperature (which then also should be used in the buoyancy terms). Finally, the heat flux that appears in equation (4) is the subgrid heat flux: (1) apparently you denote subgrid variations by a single prime and the filtering operation by an overbar and (2) in the model the subgrid heat flux is parameterized using a gradient hypothesis (also the next term, the transport term, is parameterized).

We have updated the equation adopting the convention followed by Maronga et al. (2015). The moisture tendency equation is added and the virtual potential temperature is included in the buoyancy term. We now denote the sub-grid heat flux with double prime and have also added the parameterization by gradient hypothesis. All the symbols are listed in Table 1.

7. 4-20: ‘guarantees a stable’: how does the choice of the time integration method guarantee a stable solution. The magnitude of the time step would still play a role (and it does, as later on you invoke the CFL criterion). So why mention stability here?

The combination of Runge-Kutta-2 integration and the $5^{th}$ order advection scheme is known to be conditionally unstable. The default time integration and advection scheme in PALM are RK3 and $5^{th}$ order upwind discretization according to Wicker and Skamarock, respectively.

Modified the sentence as:
“The low storage RK3 scheme with three sub-steps proposed by Williamson (1980) guarantees a stable numerical solution in combination with both the advection schemes”.

8. 4-23: I assume that you refer the vertical zero pressure gradient here.
Yes, we refer to vertical zero pressure gradient. Updated the text as:
“A vertical zero pressure gradient at the surface guarantees the vertical velocity to be zero.”

9. 5-9: apart from updating the ghostpoint, there is also global communication needed in the Poisson solver. This involves way more communication than the ghostpoint update.
We agree that the global communications for the Poisson solver need more communication than the ghost point update. Updated the sentence as:

"The data exchange between PEs needed by the Poisson solver and to update the ghost points are performed via the Message Passing Interface (MPI) communication routines."

10. 5-10: regarding the structure of the rest of section 2: I would suggest to restructure this section as follows:
2.2 Model structure
2.2.1 Grid configuration (now 2.2, up to line 5-29)
2.2.2 Nesting algorithm
2.3 Translation between grids (line 5-9 until7-4)
2.3.1 Anterpolation
2.3.2 Interpolation
2.4 Parallel inter grid communication
(after 2.2.2 it is clear where and why anterpolation and interpolation are needed).

The section 2 is re-structured as suggested.

11. 5-30: only the vertical velocity really has a boundary at the top of the FG. For the other velocity components and scalars it is unclear whether the boundary condition (interpolation from CG) is applied to a ghost point (just above the FG) or to the first point just below the boundary of the FG.

The boundary condition is applied at the boundary level. This level is excluded from the CFD computations and only acts as a boundary constraint in the CFD equations for the neighbouring prognostic grid cells. We prefer to call it as boundary point instead of ghost point and keep the terminology of ghost points for the grid points which are constraints on one parallel processing element but CFD point on another processing element. The (vertical) boundary level is a boundary level on all processing elements. We added the following sentence in section 2.3.1:

"We define the top of the FG as the boundary level just above the prognostic level of each quantity."

12. 5-32: what is the 'logical' direction? If figure 1 would be upgraded (see below), this 'logical' linear interpolation would probably become clear.
The logical direction is the dimension corresponding to the velocity component. We replace it by "in its own dimension" in the manuscript:

"For the velocity components, the interpolation is linear in its own dimension, and quadratic in the other two directions."

13. Figure 1: the current figure is not very informative. I would suggest to replace it by a figure in which you show a few CG cells as well as the FG cells within one or two of them (preferably with a grid ratio of 3, not more). Then clearly show how the interpolation of vertical velocities, as well as horizontal velocities and scalars works (in order to support the interpretation of equations (5) as well as the notion that the velocities are interpolated in a 'logical' direction. The connection to equations (5) could also clarify the meaning of the various indices (lowercase and uppercase).

As suggested we have included a schematic of a nested grid with a nesting ratio of 3. However, we would also like to retain the original figure as it could be informative to readers not familiar with the nesting procedure. The explanation to the equations have also been expanded.

[Figure]

Figure 1. (a) Schematic of the interpolation and anterpolation between grids. The FG top boundary condition is interpolated from the CG. The CG prognostic quantities in the overlapping region are anterpolated from the FG. (b) Schematic of Arakawa C grid for two grids with nesting ration of three. The black arrows and circles are CG velocity and pressure, respectively. The blue and red arrows are horizontal and vertical velocity, respectively, in the FG. The filled black circle is the FG pressure. The symbols $\Phi$ and $\phi$ represent CG and FG scalar quantities. Where I and K are CG indices and nx and nz are the nesting ratio in x and z, respectively.

14. Please completely rework the equations and add explanations:
- Make clear that the first equation is the actual interpolation, and all the other equations just define the various parameters occurring therein.

We have re-numbered the equations to make the distinction. We kept the indices for the anterpolation equations, but we changed the indices in the interpolation, to make it less confusing.

- In which coordinate direction does i vary: only in the x-direction, or also in other directions. Or are we actually looking at a 2D or 3D stencil of which only one dimension is shown?

i varies only in x. Yes, we are only looking at one dimension of the stencil (added in the text). For the interpolation, we renamed this index to "m" because it has a different flavour than the index "i" in the anterpolation.

- The capital indices I, J, and K are counting through the entire domain, I assume. But how about the lower case indices: do they start counting at 1 (or zero) within each CG cell, or do they also count globally?

We understand that the reviewer is referring to the anterpolation equation. The lower case indices only count over the fine grid cells that belong to that particular coarse cell. So for each (I,J,K) tuple it is a restricted set of (i,j,k). However, due to the grid conventions in PALM the (i,j,k) have global numbers. Due to the setup of the grid indices in PALM, this is also true for the parallelization with MPI. However, whether the i,j,k are defined locally or globally is a matter of how the nesting is applied practically, and it does not influence the nesting philosophy.

In the text: "The lower case indices only count over the fine grid cells that belong to that particular coarse

cell. For each (I,J,K) tuple there exists a restricted set of (i,j,k) indices in the FG. In order that the nested PALM knows at all times which fine grid cells and coarse grid cells correspond, we compute a mapping for the FG and CG indices before starting the simulation, and we store this mapping in the memory."

For the interpolation equation, I is global again but (to make the equation better readable) m is local, running from 1 to nx.

- In the 2 nd -4 th equations you introduce H k . What is the value of the index k. Or does the repeated index imply summation? If so, what is the range of values that k can take: 1, 2 and 3 because of the dimensionality, or 1, 2, ... n x because of the number of FG cells in a CG cell?

We admit that the k is confusing here and we have replaced it by m as well (in the original document i and k should have been equal). There is no summation convention here (added in the text). We added some lines explaining the philosophy behind the interpolation equation.

15. 6 - equation (6): what is the range of values for i, j and k ? Is there a mapping that gives the global i,j,k values for a given I,J,K or, are these local i,j,k values, running as 1, 2, ... n x ?

Yes, there is a mapping. This mapping is essential for the nesting algorithm to match the corresponding cells in the fine and coarse grid, and it is computed in advance and stored in the memory. These i,j,k values are not global.

Added in the text (see also above): "In order that the nested PALM knows at all times which fine grid cells and coarse grid cells correspond, we compute a mapping for the FG and CG indices before starting the simulation, and we store this mapping in the memory."

16. 7 - equation (7): idem
Equation for velocity anterpolation has also been updated similar to the equation for scalars.

17. 8-7 and 11: please keep the discussion on the solution of the Poisson equation in one place.
**What is the value of the pressure gradient that is imposed as a Neumann boundary condition? If it is zero, reflections could occur**, but if you use something non-zero: how do you determine the value of this gradient? Is it derived from the CG pressure field?

We have re-arranged the sentences to keep the discussion of Poisson equation in one place.

We use a zero gradient Neumann condition for pressure. We would like to quote the opinion of R1 here: "In this model the nested upper boundary condition is relatively simple, which is OK for the Boussinesq LES problem presented here in which there are no sound waves and any vertically-propagating gravity waves would be very well-resolved.". In what we refer to as the 'simplified sponge-layer', we have also split the level of the FG upper BC from the highest level of anterpolation of FG to CG in order to reduce oscillations originating from the FG boundary.

18. 8-12 and 13: please clarify how the value of the imposed pressure gradient is determined/chosen.
We impose zero gradient at the top and bottom of the nested domain.

19. 8-31: '... the higher number of PE available in the FG.': this is stated as if the reader already knows that there are more PEs in the FG (although for any grid ratio above 2 it is indeed logical that the number of FG PEs is larger than the number of CG PEs). But in addition, itis unclear to me why the higher number of FG PEs would be relevant for the FG-to-CG communication.

To perform the anterpolation operation, either the FG data can be sent to the CG and then be anterpolated, or alternatively, and more efficiently, the anterpolation operation is performed in the FG and then the anterpolated values are sent to the CG. The latter approach benefits from the higher number of FG PE and smaller array dimensions of the anterpolated values.

We have modified the sentence as:
"The exchange of arrays via MPI_SENDRECV routines is computationally expensive. Therefore, the size of the arrays communicated are minimized by performing the anterpolation operation in the FG PE's and storing the values in a temporary 3D array that is later sent via the global communicator to the appropriate CG PE. This approach is more efficient than performing the anterpolation operation on the CG which has less PE's and needs communication of larger arrays from the FG."

20. 9-2 'should be kept lower': please explain the logic of this statement. I assume that the idea is that you want to reduce the total amount of idle CPU time on the FG PEs (N cores x wait time), which can be achieved by under-utilization of the (only) M cores running CG (better waste time on a few CG cores than on many FG cores). In order to know how this plays out in practice, you should show in your results the amount of time spent in the various steps in a RK substep: which fraction (and absolute time) of a time step is devoted to which substep in figure 2, and how much of this time is wasted time.

The work load of CG PEs are kept lower than the FG PE to reduce the total amount of idle CPU time on the FG PEs. Unfortunately, in the nested simulation the time spent in various steps is not profiled. We will not be extending our current parallel implementation as the main developers of PALM will be developing the 'PALM Model Coupler', a unified tool to handle the communication between the grids for nested simulations, ocean-atmosphere coupling etc.

"Within the RK3 sub-steps, when one grid executes the pressure solver the other grid has to wait leading to more computational time at every sub-step. However, the waiting time can be minimized by effective load balancing, i.e. the number of grid points per PE in the CG should be kept lower than in the FG. The reduction in workload per CG PE is achieved with a few additional cores. The reduction in computational time per step in the CG means the idle wait time on the FG PE is also reduced."

21. 9-12 'Dirichlet condition': to which values are the velocities set: zero for vertical wind and geostrophic for horizontal?
We have updated the text as
"The Dirichlet boundary condition is applied for velocity at the top and bottom boundaries, the vertical velocity component is set to zero and the horizontal components are set to geostrophic wind."

22. 9-13: what is the imposed temperature gradient at the surface?
Since we prescribe a constant surface heat flux, we use zero gradient Neumann condition for the potential temperature. Updated the text as:

"The potential temperature is set to Neumann condition at the bottom and the gradient is determined by MOST based on the prescribed surface heat flux and roughness length. The gradient of the initial profile is maintained at the top boundary."

23. 9-15 and 16: is the wind profile interpolated linearly from zero at the surface to geostrophic at the top? Does this out-of-balance initialization lead to an inertial oscillation?

Earlier we had written "The u and v profiles are constructed starting from a zero value at the surface and reaches the geostrophic wind value at the top."

We correct the statement as:
"The u and v initial profiles are set to be constant value of the geostrophic wind component in the domain and the vertical velocity is initialized to zero in the domain. "

The initial profiles are set to be constant (ug = 1, vg = 0). Whilst it is true that the initialization is out-of-balance, the amplitude is small with respect to the surface heating in our convective boundary layer. Observing the time series of total kinetic energy of the flow (3D domain average) shown in the figure below, we can see that the oscillations subside after 1 hour of spin up. Similarly, plotting the time series of the absolute maximum vertical velocity, we observe that the maxima is almost constant after 1 hour spin up phase.

[Figure]

24. Table 1: what is the boundary condition for wind? MOST with an imposed roughness length (what is the value) or an imposed stress?

It is MOST with an imposed roughness length. Updated Table 2 to list roughness length value as 0.1 m.

25. Table 2:
- please add the number of time steps needed to complete this simulation (in that way the reader can easily determine the time spent per gridpoint per time step.
We have added a column for time steps.
- For the reader it would also helpful to include the number of grid points per PE and the CPU time per grid point (but both numbers can be derived from the available data, so the reader could do it for themselves).
We have added a column for the number of grid per PE, this column indeed readily informs the reader that FG and SA-C have same workload per PE. However, we have not included he CPU time per grid point.

- the number of cores devoted to CG and FG respectively is not motivated. Whereas later on

you advise to assign to a CG PE 40-80% of the number of grid points that is assigned to a FG PE, here you use a fraction of 16%.

We used a machine with 20 cores per node and allotted all the cores in one node to the CG. Even though it is possible for the CG and FG to share a node, the domain decomposition restrictions often prevent an ideal grid configuration. The limitations in domain decomposition are now included in the practical considerations section 3.4:

"For better computational performance it is recommended that the number of grid points per PE in the CG is kept at only 40 to 80 percent of the FG value. The reduced work load of the CG is expected to minimize the waiting time of the FG during the concurrent time advancement by quicker CG pressure solver step. However, the actual improvement in performance will depend on the memory available, processor speed and the inter-node communication architecture of the computing cluster and the optimal load balancing can only be found through trials. Furthermore, the choice of the domain size is often restricted by the topology of the processor decomposition. In a 2D decomposition, the number of grid points along the x-direction should be an integer multiple of the number of PE along x and similarly for y-direction. This condition has to be individually satisfied for the CG and the FG."

- it is unclear to what extent the PE's are saturated in terms of memory usage: could this problem be run on even a smaller number of processors to improve performance?
  We have two simulation set-ups: one simulation is performed on a small cluster to demonstrate the quality of the results and  the other simulation is performed on a supercomputer to demonstrate the computational performance. However, the first simulation is still relevant in understanding the effect of grid resolution on time step and the associated increase in the core-hours.
- Please include information on the time (absolute and/or as a fraction) that is used waiting for input from CG to FG or the other way around. This would be helpful to determine the optimal division of labor between CG PEs and FG PEs (in terms of grid points per node).

In our benchmark runs, we did not profile the time taken by each nesting routine, and therefore a detailed analysis of the overhead is not possible.

26. 10-2: what initial perturbation is applied to get turbulence started? How did you verify that after 9000 s the flow was in equilibrium?

Random perturbations are imposed to the horizontal velocity field. If the perturbation energy has exceeded this energy limit of 0.01 m²/s², no more random perturbations are assigned.

Observing the time series of total kinetic energy of the flow (3D domain average) shown above in response to question - 23, we can see that the flow has reached a quasi-stationary state after 1 hour of spin up. Similarly, plotting the time series of the absolute maximum vertical velocity, we observe that the maxima is almost constant after 1 hour spin up phase.

27. 11-5: part of the ingredients for the scaling variables are in fact imposed boundary conditions (the surface heat flux), whereas indeed another part (the surface shear stress) results from the flow (and hence need to be derived from one of the simulation results (assuming that a roughness length is prescribed).

We have replaced $u_*$ with  $w_*$ for the normalization of velocity variance as suggested in detailed comments - 31. While we acknowledge that the u* and w* result from the flow, it is more common to

normalize the velocity variance with these scaling variables instead of imposed flow variables like the geostrophic wind, because the latter is not a scaling variable in the ABL.

28. 11-6 and 8: surface heat flux in the expression for w*: overbar is missing and this is not a turbulent flux (so do not use a covariance flux).

We now represent the surface heat flux with the symbol $H_s$.

29. 11-8: although it will not change the lines in the graph, normalizing the temperature with the surface value is very illogical. Please plot the temperature with some reference value (e.g. the surface value) subtracted and normalized with θ*.
We have updated the figure to plot the temperature with surface value subtracted and normalized with $\theta_*$.

30. 12-1: what would/could be the mechanism that makes that the higher resolution in the surface layer would affect the variance profile well above the FG domain?
We realized that in the initial runs we had failed to include the compiler flag (-fpmodel strict) necessary to make every realization deterministic. As a result the plots in our initial submission have compiler optimization related random effects that were noticeable in the variance profile. The results of both the nested and the standalone simulation were affected. We have now ensured that the simulations are deterministic, which improves their comparability. The plots and their description have been updated accordingly.

In the updated plots, the CG profile is not noticeably affected by the higher resolution in the surface layer. Therefore, we have removed the statement.

31. 12-3: please use the same scaling variable for all velocity components! If not, the different variances (which together constitute the turbulent kinetic energy) cannot be compared. Furthermore, the given flow is close to free convection, so using the friction velocity as a scaling variable does not make sense.
The velocity variances are now normalized only by the convective velocity scale. The figures have been updated

32. 12-5: you refer to an overshoot in the v variance. The u variance shows an overshoot as well.
I assume that the profiles shown are based on the resolved variances only. In that case, we should keep in mind that in the CG domain a larger proportion of the TKE is contained in the subgrid scales. Could this explain the jump? **Please include an analysis of the difference in SGS-TKE between the two domains at the top of the FG (of course there is the difficulty of separating the SGS TKE into the three components, but at least quasi-quantitatively such an analysis could shed light on these jumps/overshoots.**
The plots in our initial submission had random errors introduced due to wrong choice of compiler optimization flags that was visible in the v-variance. However, there are still minor artifacts

We agree that the jump could be explained by the large SGS TKE component in the CG. We updated the text as:

"At the nesting height the variances deviate more from the SA-F and approach the CG values. Due to conservation of total kinetic energy across the nest boundary more CG TKE is contained in the sub-grid scale. Consequently, the resolved CG variances could have an undershoot as compared to SA-F,

resulting in an undershoot of the FG variances too at the nesting height. Above the nesting height, the variance of u, v and w in CG are similar to SA-C."

33. 12-7: how would the anterpolation influence the vertical velocity variance in the FG domain. Please explain the/a mechanism. Or is it a result of the fact that the upper boundary conditions for pressure at the top of the FG is not well-defined?

Zhou et al. (2018) in the vertically nested LES note that the kink in the higher-order profiles can be minimized by increasing the depth of the sponge layer. In our two-way nesting we have used a simplified sponge layer by limiting the anterpolation to one CG cell less than the nested height. This split in the level of the FG upper BC from the highest level of anterpolation of FG to CG reduces oscillations originating from the FG boundary.

In the description of the skewness plots we write:
"However, at the nesting height a small kink in the skewness is noticeable. Zhou et al. (2018) observe that the magnitude of the kink in the higher-order profiles can be minimized by increasing the depth of the sponge layer. Our simplified sponge layer approach appears to be unable to effectively minimize the kinks at the nesting height."

34. 13 figure 6: the variance profiles give some information on the quality/realism of the simulated turbulence. One analysis that is missing (related to the point made above regarding the overshoot) is whether the increased resolved TKE is the amount that would be expected based on the increased resolution (and hence reduced reliance on the subgrid model). To properly analyse that one would need turbulent spectra to see how much kinetic energy is contained in the additionally resolved scales. Additionally, spectral analysis (preferably with 2D spectra) would help to show to what extent the extra resolved turbulence has the expected turbulent characteristics (increased variance is nice, but does not need to be additional turbulence, it could also be increased noise).

We have included a spectral analysis. The plots are described as:

"The horizontal spectra of SGS turbulent kinetic energy and vertical velocity are plotted in Fig. 9 at two levels, one within the nested grid and one above the nested grid height. The FG TKE spectra in Fig. 9 (c) perfectly overlaps the SA-F spectra. The CG spectra has higher energy than the SA-C, this corresponds to the higher CG TKE values observed in Fig. 8 (c). As the limit of the grid resolution is reached at high wavenumber, the drop in the CG spectra is marginally shifted compared to SA-C. This improvement at high-wavenumber is due to feedback from the FG. Similarly, the vertical velocity spectra in Fig. 8 (d) shows marginal improvement at high wavenumber for the CG with respect to SA-C. While the FG agrees with SA-C at high-wavenumber and at the spectra peak, at low wavenumber FG follows the CG spectra. At the level above the nested grid, the CG spectra agrees with SA-C for both TKE and the vertical velocity."

35. 13-1: The heat flux profile is not the prime quantity at all! For a quasi-stationary convective boundary layer with imposed surface flux the heat flux profile is the most boring part of the simulation. Provided that the entrainment flux is represented well, the flux profile is by definition linear, varying between the imposed surface flux (so no surprises there) and the entrainment flux (which, admittedly, needs to be represented correctly by the simulation:
still some freedom there). This linear flux profile is completely independent of the quality and resolution of the simulation. The only freedom there is is which part of that flux is carried by the

resolved scales and which part is carried by the subgrid model. Hence the perfect correspondence between all simulations (full FG, full CG, nested CG and nested FG). Hence, please do not use the heat flux profile as a measure of the quality of the simulation.

We agree that heat flux profile is not the best measure for the quality of a simulation. However, we would like to retain the plots because we are interested in the heat flux for other applications in our working group to study the energy balance closure in the surface layer.

36. 14-4: '...we increase the resolution further': do you mean to increase the grid ratio, the size of the FG region, or the overall resolution of the CG domain?
We have replaced the statement with quantitative analysis using the updated time step information in Table 3. The manuscript is updated as:

"The computational resources used in the simulations discussed above are listed in Table 3. The resources needed by SA-C is only 8 core hours. While the nested simulations needed about 1879 core hours, the SA-F needed about 4 times more core hours. As the resolution is increased from 20 m in SA-C to 4 m in SA-F the number of time steps increased more than 5 times as higher resolution demands smaller time step size."

37. 14-9: 'in terms of communication time': do you only look at communication time because that is the most restricting, or because you are only interested in that (in this context)? And why should the number of domains be equal in x and y direction: please explain the logic of this (and does it also hold if the length of the domain is different in x and y direction?

We are interested in the communication time in the context of the domain decomposition because, the choice of processor decomposition has an effect on the communication performance. We update the text as:
"The best performance in terms of communication time in a standalone run is achieved when the number of sub-domains in the x and y directions are equal. In that case the number of ghost points at the lateral boundaries are optimally minimized."

38. 15-1 to 5: why is the setup of these simulations (in terms of the total number of points and ratio of number of grid points between CG cells and FG cells) so different from the original runs? Are the performance results still relevant to understand those first runs? If so, why?
Please give the setup of these runs in a table similar to table 2 (not 'number of grid points is around...').
We have two simulation set-ups: one simulation is performed on a small cluster to demonstrate the quality of the results, and another simulation is performed on a supercomputer to demonstrate the computational performance. The first simulation is relevant in understanding the effect of the grid resolution on the time step and the associated increase in core-hours. The second set of simulations demonstrates the scalability on large number of cores. Since both machines have different processor architecture, memory and processor per node, the results are not directly comparable.

In the performance benchmark set-up CG and FG PE's are chosen to be a multiple of 16 to confirm with the 16 cores per node. To avoid load balancing bias in the scalability analysis, the ratio between the number of PEs for CG and FG is kept constant in all the five runs listed in Table 5. Keeping the processor ratio constant implies that the ratio between the number of grid points per PE in CG and FG is also held constant. Consequently, in this performance test, the FG has 1.25 times more grid points per PE than the CG in all the processor configurations tested.

We ave added two tables: "Table 4. Number of grid points in nested and non-nested FG domain." And "Table 5. Grid configuration of the nested and non-nested FG domain."

39. 15- Figure 8:

- on a log-log scale everything looks nice. Please give a more informative representation. E.g. use the strong scaling efficiency, which will vary between 1 and somewhere below 1 (for your data, using the left-most simulation as a reference, the efficiency goes down to about 90% for the right-most. But the question is, what would have been the CPU time for the smallest possible number of processors on which this case could have been run (memory-wise).

We computed the strong scaling efficiency as suggested and included the information in Table 5. We have only tested on total PE above 1664. However, for the smallest processor configuration, the FG group has 3.9 million grid points per PE when the usable memory on this machine is only 1.625 GB per PE.

- In addition, find a more informative way to quantify the waiting time overhead.

In our benchmark runs, we did not profile the time taken by each nesting routine, and therefore a detailed analysis of the overhead is not possible. To overcome this limitation in our analysis, we executed standalone simulations with the same number of grid points as in the FG domain. The difference between the nested and the standalone in the scalability plot provides a rough estimate of the overhead.

40. 16-9: 'large scale forcing .... compatible': do you refer to the large scale forcing in terms of pressure gradient/geostrophic wind? Or another large scale forcing? Why would it, or would it not, be compatible. Please clarify.

We refer to the large scale forcing feature in PALM, which is PALM driven by a.o. geostrophic wind from a gridpoint of a synoptic model (see Maronga 2015 for more details). We did not expect it to be incompatible but in model development it is not guaranteed that two different modules are directly compatible, so we made some separate tests of the large scale forcing feature with the nesting. We deem this is useful information for PALM users who would like to combine the nesting module with other modules in PALM.

41. 16-12: 'accuracy': for accuracy in what sense (interpolation errors, truncation errors, turbulence statistics, stability, ....) should the grid ratio be not too large?

We expect that too large nesting ratio would affect the accuracy of the turbulence statistics. Since we do not have extensive analysis of larger grid ratios we have removed the sentence.

42. 16-14: 'first five grid points are unreliable': for which variables does this hold, in which aspect are the grid points unreliable (I assume you mean 'the results at the first five grid points vertically displaced from the surface'): turbulence characteristics, mean profiles, noise, ....? Do you have a reference for this bold statement?

We meant the first five grid layers in the vertical direction form the surface are unreliable because the resolved turbulence is not well developed yet. This is only based on our experience; due to lack of literature to support this point we have removed this sentence.

43. 16-16: I would like to see a quantitative motivation for this 40-80%.
Unfortunately we do not have a quantitative analysis to support this statement. However, this is based on our trials on two different architectures, one a small computing cluster and the other a supercomputer. Since multiple factors like the memory available, processor speed and the communication architecture of the high performance machine a generally applicable quantitative analysis is difficult. However, we have expanded the practical considerations in section 3.4 to mention these factors:

"For better computational performance it is recommended that the number of grid points per PE in the CG is kept at only 40 to 80 percent of the FG value. The reduced work load of the CG is expected to minimize the waiting time of the FG during the concurrent time advancement by quicker CG pressure solver step. However, the actual improvement in performance will depend on the memory available, processor speed and the inter-node communication architecture of the computing cluster and the optimal load balancing can only be found through trials."

**Very detailed comments**

1. 2-14: 'possible, by the time' → 'possible. By the time'
Corrected as: "As computing power progressed, higher resolution and larger domains became possible. By the time of Schmidt and Schumann (1989)"

2. 2-18: 'supercomputers': also the people before Kröniger et al. used supercomputers. So remove 'with the help of supercomputers'.
Corrected as: "More recently, Kröniger et al. (2018) used"

3. 2-19: remove 'speeds'
Corrected as: "to study the influence of the wind on"

4. 2-21: 'higher detail': 'higher' than what/when/who?
Corrected as: "The atmospheric boundary-layer community has greatly benefited from the higher spatial resolution available in these LES to study turbulent processes that cannot be obtained in field measurements"

5. 2-28: 'Nesting has been applied...': because the previous sentence talks about vertical nesting, the reader may think that this sentence gives examples of that. But then at the end it turns out to talk about horizontal nesting. Please rephrase.
The sentences have been re-arranged following detailed comments 2.

6. 3-3/4: 'techniques are ..... but often uses ...': 'uses' should be 'use' (plural)
Corrected as: "but often use different terminology."

7. 3-18: '... CG, there ...' → '... CG, and there ...'
Corrected as: "FG receives information from the CG, and there is no feedback to the CG"

8. 3-24: make explicit that the 'two different approaches' only refer to the 'anterpolation' mentioned in the sentence before. Furthermore, nothing is said –explicitly- about the 'pressure deficit correction' ("there are two types of cars: blue cars")
Only the post-insertion approach involves anterpolation. We have incorrectly spelled 'pressure defect correction' as deficit. Updated the text to clarify on 'pressure defect correction':

"In the post-insertion technique, once the Poisson equation for pressure is solved in the FG, the resolved fields are then anterpolated to the CG. In the pressure defect correction approach, the pressure in the CG and FG are matched by adding a correction term to the CG momentum equations and an anterpolation operation is not required."

9. 4-9: 'additional equation': additional to what? The SGS-TKE equation is in the Deardorff method, so it is not additional to his work.
Corrected as: "The sub-grid scale (SGS) turbulence is modelled based on the method proposed by Deardorff (1980)."

10. 4-11: 'The prognostic ...': move this sentence to below the equations (only after having presented the equations you need to talk about their discretisation).
Corrected as: "The prognostic equations for the resolved quantities are:"

11. 5-17: 'the grids' → 'grids' (this occurs in multiple places, please check.
Correct 'the grids' to 'grids' in all the occurrences.

12. 5-18: please explain here already that uppercase symbols refer to CG and lowercase symbols to FG.
Moved the sentence in front of the first use of the symbol: "Below we use upper case symbols for fields and variables in the CG, and lower case for the FG. "

13. 6-1: 'similar interpolation': in which way is it similar, and which way is it different?
Corrected as: "The same interpolation formulation is also used to initialize all vertical levels of the fine grid domain at the beginning of the nested simulation."

14. 6-5: 'scalars' → 'CG scalars'
The statement is general to the Arakawa C grid and not specific to the CG. We modified the sentence as: "In the Arakawa C-grid discretization that PALM uses, the scalars are defined as the spatial average over the whole grid cell"

15. 6-5: 'The scalars .... corresponding FG scalars (eq. 6)'. How much more are you saying than 'An average is an average'. If you want to state more, please make that clear and explicit.
Modified the sentence as clarified in the comment above:
"In the Arakawa C-grid discretization that PALM 5 uses, the scalars are defined as the spatial average over the whole grid cell, and therefore it is required that the CG scalar is the average of the corresponding FG scalars in (Eq. 15)."

16. 7-6: 'We implement....'. Well, that does not really come as a surprise: you gave that away already (see my suggestions for an alternative structure for section 2).
Restructured as suggested in detailed comment 10.

17. 8-7: '... is also updated...'. What else is updated? You mean the pressure? And is the vertical velocity updated throughout the FG, or are you only referring to the vertical velocity at the CG-FG interface?
The pressure solver along with the pressure also updates the vertical velocity. The pressure solver is updates entire FG domain. Updated the manuscript as:

"The Poisson equation is then solved for pressure in the FG and the vertical velocity in the FG is also updated by the pressure solver at this stage."

18. 8-20: '...process. Whereas....' → '...process, whereas....'
Corrected as: "The data between the processors of the same group are exchanged via the local communicator created during the splitting process, whereas the data between the two groups are exchanged via the global communicator exchanged via the global communicator"

19. 8-20: 'exchange' → 'exchanged'
Corrected as: "The data between the processors of the same group are exchanged via the local communicator"

20. 8-23: 'local PE's 2D processor co-ordinate': in what way is the PE different from the processor? → 'local 2D processor (or PE) coordinate'
The terms processor and processing element are often used interchangeable. However, a subtle difference exists. While a processor is considered as a hardware unit, a processing element is a MPI task (a Unix process) that executes the program on a unique sub-set of data.

21. 9-11: 'is set to' → 'has'
Corrected as: "The simulation domain has periodic boundary conditions"

22. 10, Tables 1 and 2: please format the tables properly as tables should be formatted (including column headings and a consistent demarcation of rows and columns)
Table 1: please note that the surface heat flux is not a turbulent flux (there is no vertical velocity (variation) at the surface. Furthermore, even if you would like to denote it as a turbulent flux, please add an overbar.
Format of all the tables have been updated. We adopts the symbol Hs for surface heat flux.

23. 11, figure 4: the lower panel is –vertically- not exactly to scale with the area indicated with the dashed line in the upper panel.
We have resized the subplots to scale.

24. 11-2: 'flux profiles' → 'fluxes'
Corrected as: "The turbulent fluxes are obtained using the spatial covariance"

25. 11-3: the given expressions are not fluxes, but products of resolved deviations: please include an averaging operator to make it a flux.
We have included an averaging operator.

26. 11-13: 'at the boundary layer height' → 'at the top of the boundary layer'
The sentence has been removed as the plot has been updated.

27. 12-1: 'An one-way' → 'A one-way'
The sentence has been removed as the plot has been updated.

28. 12-5: 'variance seen' → 'variance can be seen'.
The sentence has been removed as the plot has been updated.

29. 14-5: 'Simulations with O(1) ...': are you referring to that resolution for a full domain, or

only for the FG part of a nested simulation? In fact, it is unclear where you are heading with lines 14-2 to 14-6.
We have replaced the statement with quantitative analysis using the updated time step information in Table 3. As answered in detailed comments 36.

30. 14-13: 'new nested simulation': new relative to? I assume that you mean new relative to the runs described in tables 1 and 2. These new simulations were made for the performance test only?
The new simulations were made only for testing performance. Updated the text as:
"Keeping these factors 5 in mind, we designed the nested simulation domains listed in Table 4 for the purpose of assessing the computational performance as the total number of processors is varied."

31. 16-4: 'Poisson equation' → 'the Poisson equation'
Corrected as: "Similarly, for solving the Poisson equation for the pressure"

32. 16-7: 'FFT' → 'an FFT'
Corrected as: "therefore an FFT based pressure solver is an appropriate choice"

33. 16-28: 'energy conserving methods': I have not seen that term earlier in the paper. Where was this discussed before? Or are you referring to the anterpolation of SGS TKE? In that case, please be a bit more explicit.
We are referring to the anterpolation of SGS TKE. The text is updated as:

"The grids are coupled, i.e the interpolation of the boundary conditions and the feedback to the parent grid are performed, at every sub-step of the Runge-Kutta time integration. The anterpolation of the TKE involves the Germano identity to ensure the conservation of total kinetic energy."

34. '... optimized for performance': how were they optimized, where can I read about that optimziation?

We have updated the sentence as:
"The exchange of data between the two grids is achieved by MPI communication routines and the communication is optimized by derived datatypes."